# Engineered allogeneic T cells decoupling T-cell-receptor and CD3 signalling enhance the antitumour activity of bispecific antibodies

Edo Kapetanovic [1], Cédric R. Weber[1], Marine Bruand [2,3,4], Daniel Pöschl [1], Jakub Kucharczyk[1], Elisabeth Hirth[1], Claudius Dietsche [1], Riyaz Khan[1], Bastian Wagner[1], Olivier Belli [1], Rodrigo Vazquez-Lombardi[1], Rocío Castellanos-Rueda [1,5], Raphael B. Di Roberto [1], Kevin Kalinka[1], Luca Raess [1], Kevin Ly [1], Shivam Rai[1], Petra S. Dittrich [1], Randall J. Platt [1], Elisa Oricchio[2,3,4] & Sai T. Reddy [1]✉

Bispecific antibodies (biAbs) used in cancer immunotherapies rely on functional autologous T cells, which are often damaged and depleted in patients with haematological malignancies and in other immunocompromised patients. The adoptive transfer of allogeneic T cells from healthy donors can enhance the efficacy of biAbs, but donor T cells binding to host-cell antigens cause an unwanted alloreactive response. Here we show that allogeneic T cells engineered with a T-cell receptor that does not convert antigen binding into cluster of differentiation 3 (CD3) signalling decouples antigen-mediated T-cell activation from T-cell cytotoxicity while preserving the surface expression of the T-cell-receptor–CD3 signalling complex as well as biAb-mediated CD3 signalling and T-cell activation. In mice with CD19+ tumour xenografts, treatment with the engineered human cells in combination with blinatumomab (a clinically approved biAb) led to the recognition and clearance of tumour cells in the absence of detectable alloreactivity. Our findings support the development of immunotherapies combining biAbs and 'off-the-shelf' allogeneic T cells.

Many cancer immunotherapies rely on the functional activation of endogenous host T cells. For example, T cells engaging bispecific antibodies (biAbs) simultaneously target the cluster of differentiation 3 (CD3) molecules on T cells and a defined antigen on tumour cells, acting as molecular bridges that drive cytotoxic T-cell responses. A prominent example is blinatumomab, a clinically approved biAb used in treating haematological malignancies (for example, B-cell acute lymphoblastic leukaemias). Blinatumomab simultaneously binds to CD3 (T cells) and CD19 (a ubiquitous B-cell marker) and has successfully increased survival for patients with relapsed/refractory B-cell acute lymphoblastic leukaemia[1,2]. Unfortunately, in blinatumomab-treated patients, it is common to observe relapse[3,4] not driven by tumour antigen escape (CD19+ tumour cells are still present). While the reasons for failure are multi-faceted, the immunocompromised status of cancer patients likely plays a notable role due to a depleted fraction of healthy and functional autologous T cells[5,6]. This has been outlined in a study on

[1]Department of Biosystems Science and Engineering, ETH Zurich, Basel, Switzerland. [2]Swiss Institute for Experimental Cancer Research, Lausanne, Switzerland. [3]School of Life Sciences, EPFL, Lausanne, Switzerland. [4]Swiss Cancer Center Leman, Lausanne, Switzerland. [5]Life Science Zurich Graduate School, Systems Biology, ETH Zurich, University of Zurich, Zurich, Switzerland. ✉e-mail: sai.reddy@ethz.ch

mosunetuzumab, a clinically approved anti-CD20 biAb where it was demonstrated that a primary determinant of mosunetuzumab activity is the amount of T cells the patient has[7]. These findings were further corroborated in a recent study with teclistamab, an anti-B-cell maturation antigen bispecific, which showed that the difference between responders and non-responders is in their T-cell composition[8].

In immunotherapy, allogeneic T cells are actively being investigated (as in chimeric antigen receptor (CAR) T cells) due to their promise for ease of manufacturing, scalability and therapeutic consistency. In addition, combination immunotherapies such as adoptive T-cell transfer (CAR-T, T-cell receptor (TCR)-T cells and tumour-infiltrating T lymphocytes) combined with antibodies targeting immunomodulatory receptors (PD-1 and CTLA-4) are becoming a clinically valuable therapeutic strategy[9,10]. Similarly, a combination of biAbs with allogeneic T cells from healthy donors could provide a boost to the immune system of immunocompromised patients with cancer and thus enhance the clinical efficacy of biAb. However, such an allogeneic T-cell therapy comes with several risks. Most concerning is the human leukocyte antigen (HLA) mismatch that can induce T cell receptor (TCR)-mediated activation of donor T cells against the recipient's self-antigens leading to an adverse and severe onset of graft-versus-host disease (GvHD)[11,12]. In principle, all naturally occurring TCRs, given the appropriate HLA and peptide context, can cause an alloresponse and lead to a GvHD. Even in the efforts to achieve full HLA matching for haematopoietic stem cell transplantations, the risk of a life-threatening GvHD remains very high (50%)[13].

The TCR–CD3 complex regulates T-cell activation and response. It is a mechanosensory unit composed of the αβ-TCR dimer and CD3γε-CD3δε-CD3ζζ hexamer[14–16]. The αβ-TCR dimer is responsible for peptide-HLA recognition and transfer of mechanical forces to the non-covalently bound CD3 molecules. Antigen (peptide-HLA) binding to the TCR causes conformational changes in the receptor and triggers the CD3 signalling apparatus. CD3 molecular rearrangement activates intracellular proteins (Lck and Zap-70) and, subsequently, a cascade of cellular pathways necessary for T-cell survival, proliferation, differentiation and effector functions. The eight subunits of the TCR–CD3 complex are indispensable for functional and stable surface expression[17]. Therefore, in the context of enhancing biAbs with allogeneic T-cell transfer, it is not feasible to simply knock out TCR chains, as misfolding or absence of any of the subunits will result in a complete loss of the TCR–CD3 complex and consequently CD3 molecules that are necessary for biAb activation.

In this Article, we report the development of Allogeneic-Engineered-Decoupled (AED) T cells, which functionally decouple TCR–antigen binding from signalling while preserving natural TCR/CD3 surface expression. Through TCR germline sequence and structural analyses, we performed protein engineering on a sequence motif that was found to be highly conserved across mammalian species, including humans[18], and which was previously identified to be a flex point for TCR/pMHC (peptide–MHC) interactions[19]. By performing targeted genomic mutagenesis, functional screenings and deep sequencing in the newly discovered motif, we engineered novel TCRs that can bind their cognate pMHC and critically do not transform TCR–antigen binding into a CD3 activation signal. Furthermore, in vitro and in vivo (mouse model) studies confirmed that AED T cells can recognize and clear CD19+ human tumour cells when co-administered with blinatumomab and yet, in the presence of cognate peptide antigen, remain unresponsive, thus lowering the risk of alloreactive responses (GvHD). These findings may open new directions for designing and developing combination therapies of biAbs and healthy donor allogeneic T cells.

## Results

### TCRs with a mutation in the alpha connecting peptide motif lose the ability to respond to antigen and blinatumomab

The αβ TCR heterodimer determines T-cell specificity to pMHC complexes. TCR α and β chains consist of recombined variable regions (variable (V), diversity (D) (β chain only) and joining (J) genes) and a constant region made of three distinctive segments: a membrane-proximal connecting peptide region, a single transmembrane spanning region and a short cytoplasmic tail lacking signalling domains. CD3 molecules (γε, δε, ζζ), responsible for intracellular signalling and T-cell activation, are associated with TCRs through charged interactions in the transmembrane regions[14,20,21]. These interactions secure accurate assembly of the TCR–CD3 complex within the endoplasmic reticulum and Golgi apparatus, ensuring only a functional TCR–CD3 unit is present on the cell surface[17,22]. Disrupting expression of any of the TCR chains (such as CRISPR–Cas9 (clustered regularly interspaced short palindromic repeats and CRISPR-associated protein 9)-mediated knockout of TCR alpha chain) also results in a complete knockout of all CD3 co-receptor subunits and their signalling domains, thus rendering T cells unresponsive to both TCR- and CD3-mediated stimulation (Fig. 1a).

Previous research using murine T cells and structural components of TCR signalling revealed that mutations in the sequence motif (FETDxNLN) of the TCRα connecting peptide domain can drastically reduce (>100-fold) T-cell responsiveness to cognate antigen (pMHC) while not disrupting CD3-mediated activation[20,23] (Fig. 1b and Supplementary Fig. 1d). We first set out to investigate whether these mutations in the TCRα connecting peptide domain of human TCRs could result in a molecular decoupling of TCR and CD3 signalling. As a model cell line, we used a previously engineered human Jurkat T-cell line, which lacks endogenous TCR/CD3 complex on the cell surface and has an integrated nuclear factor of T-cell activation (NFAT)-green fluorescent protein (GFP) reporter, where GFP expression correlates with TCR–CD3-mediated activation[24]. We used CRISPR–Cas9 and homology-directed repair (HDR) to genomically integrate the complete DMF5 (MART-1 specific T-cell receptor) TCR gene into the Jurkat cell line and restore TCR–CD3 surface expression (Jkt-DMF5) (Supplementary Fig. 1a). DMF5 is a well-characterized TCR targeting a tumour-associated melanoma antigen (MART-1) and has been previously investigated in clinical trials as an anti-melanoma immunotherapy[25,26]. DMF5 is specific to three MART-1 peptides, including a synthetic derivative peptide (ELAGIGILTV (ELA)), which induces the strongest activation. In addition, we generated Jkt-DMF5 cell lines expressing the previously described mutations in the alpha connecting peptide motif (aCPM) (Jkt-DMF5$_{FATADALN}$ and Jkt-DMF5$_{GGGSGSG}$) (Fig. 1b and Supplementary Fig. 1b), which have shown a potential for decoupling TCR and CD3 signalling in mouse T-cell experiments[20,23]. We performed overnight co-culture assays with T2-antigen-presenting cells (T2-APC) pulsed with a range of ELA peptide concentrations ($10^0$ to $10^4$ nM). Furthermore, we co-cultured the Jkt-DMF5$_{FATADALN}$ and Jkt-DMF5$_{GGGSGSG}$ cells with the target-expressing Raji (CD19+) cells and a range of blinatumomab concentrations (0–12 ng ml$^{-1}$) to investigate their response upon CD3 stimulation. T-cell activation and response were quantified by GFP expression via flow cytometry (Fig. 1c,d).

Our findings with the human-derived Jurkat T cells were inconsistent with the data reported in mouse T cells[13,18]. At the highest peptide concentration, Jkt-DMF5$_{FATADALN}$ and Jkt-DMF5$_{GGGSGSG}$ showed only a 50% reduction in peptide response (Fig. 1c–e) compared with wild-type (WT) Jkt-DMF5. Furthermore, variants had a drastic reduction (up to 80%) in response to blinatumomab (Fig. 1c–e and Supplementary Fig. 1c). In addition, we observed a substantial drop in TCR–CD3 surface expression and MART1-dextramer binding (Fig. 1f). The findings (Fig. 1f) corroborate previously observed differences between murine and human TCRs, where murine TCRs have a much greater potential of stable TCR pairing and expression, even when mutations are introduced[27]. Taken together, engineering in the FETDxNLN motif did not result in a favourable antigen binding and signalling decoupling. This is not surprising, considering the complex interactions aCPM has with CD3 subunits[14]. Moreover, the FETDxNLN motif is located at the membrane-proximal nexus of the TCR/CD3 complex, and thus, structural modifications and engineering in this sequence space render

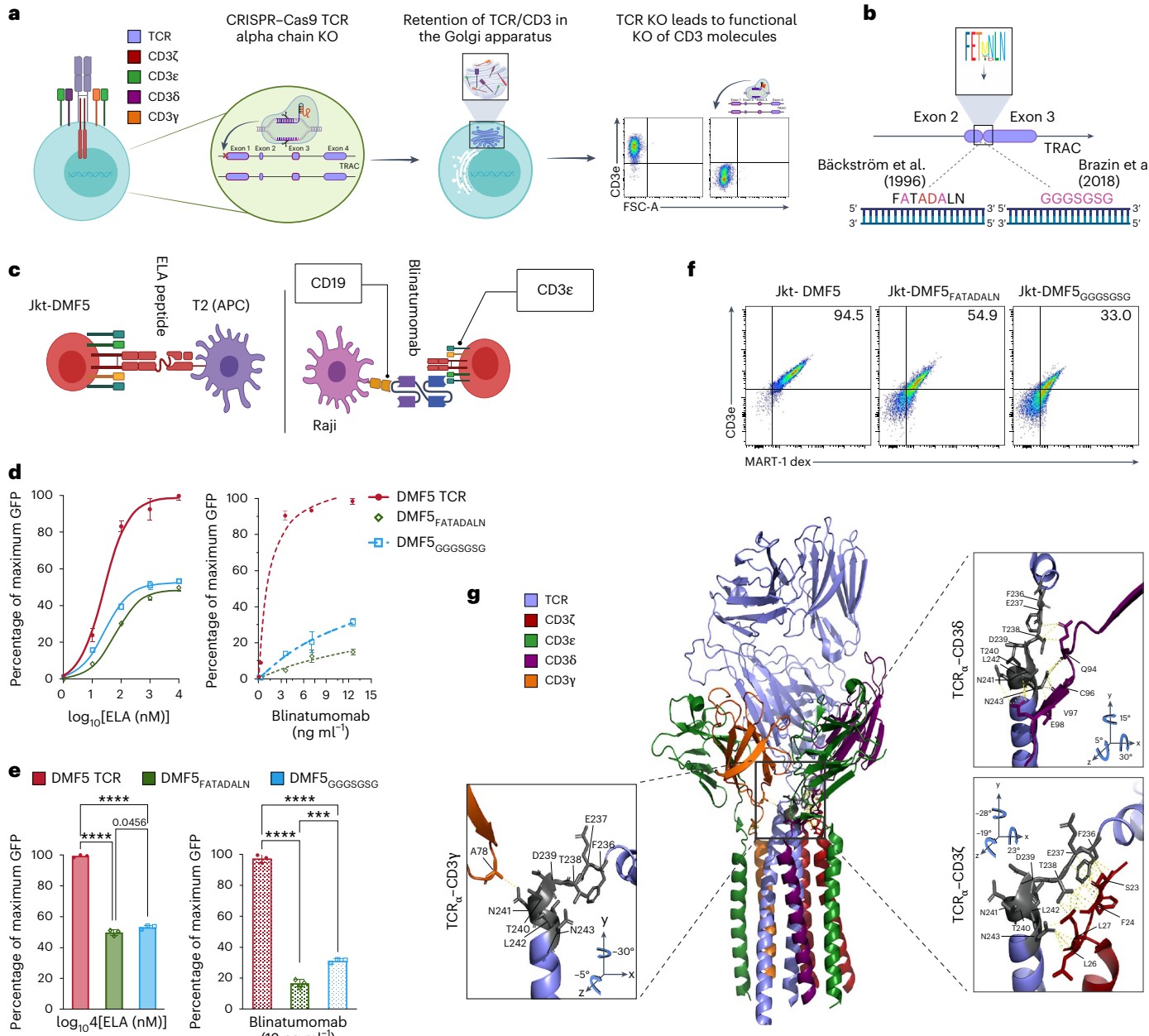

**Fig. 1 | Mutations in the aCPM of TCRs abrogate TCR–antigen binding and CD3 signalling. a**, The TCR–CD3 complex is a functionally dependent octamer; genomic knockout (KO) of any of the components disrupts a complete assembly in the endoplasmic reticulum and Golgi apparatus. Individual chains and incomplete TCR complexes are retained or degraded in the endoplasmic reticulum, and the result is loss of surface expression of the entire complex; for example, CRISPR–Cas9 KO of the TCR alpha chain constant region (TRAC) leads to a functional KO of CD3 molecules. **b**, The aCPM has been identified as a key motif for TCR–antigen (pMHC) binding and CD3 signal transduction in mouse T cells. The motif is spanning the junction of exons 2 and 3 of the TRAC locus, and its residues (FETDxNLN) are highly conserved across mammalian species. Two Jurkat cell lines expressing TCR variants with mutations in aCPM (Jkt-DMF5_FATADALN and Jkt-DMF5_GGGSGSG) are shown. **c**, Schematic representation of the co-culture assays. APCs (T2) are pulsed with a different concentration of peptide (ELA) and cultured overnight with Jkt-DMF5 cells in a 1:2 ratio; Raji cells express CD19 antigen and are cultured with blinatumomab and Jkt-DMF5 overnight

in 1:2 ratio. **d**, Left panel: overnight co-cultures with the T2 cells pulsed with MART-1 peptide antigen (ELA). Right panel: Jkt-T-cell NFAT-GFP dose response to blinatumomab (ng ml⁻¹) in co-culture with Raji (CD19⁺) tumor cells. Data are normalized to the WT Jkt-DMF5. **e**, A comparison of the highest responses of all TCR variants to a peptide (left) and blinatumomab (right) with statistical analysis. **f**, Representative flow cytometry plots of T-cell binding to MART-1 dextramer and CD3 expression in Jkt-DMF5_FATADALN and Jkt-DMF5_GGGSGSG variants. **g**, Assembly of the extracellular domains of the TCR/CD3 complex is mediated by the connecting peptides of the TCR αβ chains and their molecular interactions with CD3γε and CD3εδ. Multiple bonds between TCR-α cPM residues (F236-N243) and CD3 molecules are displayed (PDB, 6JXR). In **d** and **e**, the experiment was performed three times. Asterisks indicate statistical significance between Jkt-DMF5 variants and WT Jkt-DMF5 TCR as determined by one-way ANOVA with Tukey's post hoc test for multiple comparisons. Data are displayed as mean ± s.d. *P < 0.05; **P < 0.01; ***P < 0.001; ****P < 0.0001; NS, not significant. Significant P > 0.001 are numerically written. Panels **a**–**c** were created with BioRender.com.

TCRs unresponsive to activation from both peptides and biAbs (such as blinatumomab) (Fig. 1g).

### Structural and sequence analyses reveal a conserved TCR motif with the potential to decouple TCR–antigen binding from CD3 signalling

To engineer functionally decoupled TCRs, we first set out to identify favourable sequence motifs. The TCR/CD3 complex has multiple components, and thus introducing mutations can easily destabilize the TCR–CD3 assembly and result in the loss of cell surface expression. Simple deletion of Vα and/or Vβ domains would disrupt constant region folding, whereas misfolded domains are marked for degradation resulting in loss of both TCR and CD3 surface expression (Fig. 1a). Therefore, we devised several criteria based on sequence and desired functional properties (Fig. 2a). First, a potential TCR motif should be highly conserved across TCR germline genes. Second, it should be outside of complementarity-determining regions (CDRs) that play a key role in determining molecular specificity to cognate pMHC complexes. Amino acid substitutions in the CDRs can introduce new and unknown TCR specificity and cross-reactivity, which can have substantial consequences for the safety of a T-cell therapeutic. For example, in a previous T-cell therapy clinical trial, a TCR engineered in the CDR region for enhanced affinity to a tumour-associated antigen (MAGE-A3 peptide) showed an unexpected cross-reactivity toward a self-antigen expressed by beating cardiomyocytes; this resulted in treatment-induced patient deaths[28,29]. In addition, the motif should be outside of the aCPM, as we determined that mutations in this region result in the abrogation of both TCR and CD3 signalling in human cells (Fig. 1). Another essential criterion is that TCR and CD3 surface expression needs to be maintained. Finally, mutations in this motif must drive a loss of TCR signalling in response to cognate peptide-HLA antigen while maintaining CD3 signalling in response to agonist ligands (for example, blinatumomab).

To identify candidate motifs meeting the criteria, we performed multiple sequence alignments (MSAs) of TCR V- and J-gene germline sequences within and across species (Supplementary Fig. 2a–e). This led to the uncovering of the FGxGT motif present in the TCRα J-gene (TRAJ region); this motif is highly conserved in most human germline J-genes and across mammalian species (Fig. 2b). It is situated just outside the CDR3 region, and notably, it occurs not only in the TCRα but also in the TCRβ chain as well as in antibody heavy and light chains (Supplementary Fig. 2c). Previous work revealed that the FGxGT motif acts as a swivel, a flex point adjusting the association of Vα and Vβ domains[19]. It was discovered that the second glycine (G102) of the FGxGT motif in the J elements is the responsible residue and may affect how CDRs align and interact with individual MHC molecules. In our study, we hypothesize that the FGxGT motif could also be responsible for transmitting the signal generated by peptide binding to CD3 molecules. Thus, our engineering efforts aimed to introduce mutations into the Jkt-DMF5 receptor that would preserve TCR stability, disrupt the association of the alpha and beta chains and cease the natural signal transmission

upon pMHC engagement. As glycine residues are often found in flexible regions (and synthetic linkers) of proteins due to their small size and ability to adopt multiple conformations, to disrupt the suspected function of the motif and at the same time preserve receptor stability, we replaced the first glycine residue (G100) in the alpha chain FGxGT with glutamic acid (E), a large and negatively charged amino acid that we hypothesized would disrupt the flexibility in the region and the Vα–Vβ interchain association. Furthermore, the pivotal and second glycine (G102) of FGxGT was replaced with tryptophan (W) to stabilize the alpha chain, as aromatic residues are favoured in beta-sheet stabilization. A newly designed candidate AED T-cell variant with the mutated motif (FEQWT) was incorporated into the backbone of the DMF5 TCR and integrated via Cas9-mediated HDR into the genome of Jurkat cells (Jkt-AED$_{DMF501}$) (Supplementary Fig. 1a).

As mentioned, an essential criterion is to maintain TCR binding specificity to cognate peptide-HLA. Flow cytometry analysis of Jkt-AED$_{DMF501}$ labelled with peptide-HLA dextramers (MART-1-HLA2) revealed a binding profile indicating that Jkt-AED$_{DMF501}$ remains specific to its cognate antigen. (Fig. 2d). Next, we investigated the response of the Jkt-DMF5 and Jkt-AED$_{DMF501}$ T cells to MART-1 peptides. In these assays, we included naturally occurring peptides (AAGIGILTV (AAG) and EAAGIGILTV (EAA)) as well as the previously used synthetic peptide (ELA). T cells were co-cultured with T2-peptide-pulsed cells, and flow cytometry analysis was performed to examine the GFP response (Fig. 2e). At the highest concentration of ELA peptide (10 µM), the Jkt-AED$_{DMF501}$ cells showed more than 3-fold reduction in GFP expression. Moreover, it required a >500-fold higher peptide concentration for Jkt-AED$_{DMF501}$ to reach the same level of activation as the WT DMF5 TCR. Furthermore, Jkt-AED$_{DMF501}$ cells produced a maximum activation that was 70% lower than the WT counterpart (Fig. 2f). A similar pattern was observed across all three MART-1 peptides, with Jkt-AED$_{DMF501}$ showing virtually no activation when binding the naturally occurring peptides. It is worth noting that AED$_{DMF501}$ T cells co-cultured with Raji tumour cells (CD19$^+$) and blinatumomab were equally responsive as the WT DMF5 TCR across all concentrations tested (Fig. 2f,g and Supplementary Fig. 3c). Moreover, AED$_{DMF501}$ T cells had a comparable response when activated with anti-CD20 bispecifics (mosunetuzumab and glofitamab), showcasing the compatibility of AED T cells with various bispecific T-cell engager formats (Supplementary Fig. 3d). In addition, we examined and compared CD69 surface expression and interleukin-2 (IL-2) secretion between Jkt-AED$_{DMF501}$ and WT DMF5 TCR, and the results were consistent with the analysis of GFP expression (Supplementary Fig. 3a,b). These results provide evidence that the FGxGT motif can be engineered to decouple TCR–antigen binding from CD3 signalling. To further characterize the FEQWT (AED$_{DMF501}$) mutation, we performed the comparative modelling and molecular dynamics simulations. The analysis revealed differences in the root-mean-square fluctuation between DMF5 TCR and AED$_{DMF501}$ across all chains except CD3z. Most observable were the changes in the lipid-bilayer positioning of CD3ε and CD3ε′ chains where multiple fluctuations between WT DMF5 TCR and AED$_{DMF501}$ were higher than 50%. These observations

---

**Fig. 2 | TCR sequence analysis and functional decoupling of TCR–antigen binding from CD3 signalling. a**, Key criteria defined for identifying motifs to enable decoupling of TCR–antigen binding from CD3 signalling. **b**, Multiple sequence alignment of TRAJ germlines from humans (left) and selected mammals (right) shows a highly conserved motif (FGxGT). **c**, DMF5 TCR (PDB, 3QDJ) structure is represented with alpha-beta chain spatial conformation and colour-coded CDRs (yellow, CDR1; green, CDR2; magenta, CDR3). The magnified square depicts inter-chain molecular contacts between the α-chain FGxGT motif and β-chain residues in proximity. A complete amino acid sequence of the DMF5 TCR chains with highlighted CDRs is located in the top right corner. **d**, Representative flow plots of MART-1-HLA A2 dextramer binding. **e**, Representative flow cytometry plots of NFAT-GFP and CD3 expression in Jkt-DMF5 (red) and Jkt-AED$_{DMF501}$ cells (blue). T cells were co-cultured overnight with peptide-pulsed

T2 cells (ELA). In grey, Jkt-CD3$^-$ cell-line basal activation and in culture with the highest concentration of ELA peptide (10 µg ml$^{-1}$). **f**, Three leftmost panels: NFAT-GFP response curves for Jkt-DMF5 and Jkt-AED$_{DMF501}$ cells to three known cognate peptide antigens (ELA, AAG, EAA). Rightmost panel: CD3 activation curves with blinatumomab. Data are normalized to the Jkt-DMF5. **g**, Three leftmost panels: bar plots of the two highest peptide concentrations $4\log_{10}$(nM) (10 µg ml$^{-1}$) and $3\log_{10}$(nM) (1 µg ml$^{-1}$) measured for Jkt-DMF5 and Jkt-AED$_{DMF501}$ and their statistical analysis. Rightmost panel: two selected highest concentrations for blinatumomab. In **f** and **g**, the experiment was performed three times. Data are displayed as mean ± s.d. *P* values were determined using a two-tailed, unpaired Student's *t*-test. *$P < 0.05$; **$P < 0.01$; ***$P < 0.001$; ****$P < 0.0001$; NS, not significant. Significant $P > 0.001$ are numerically written.

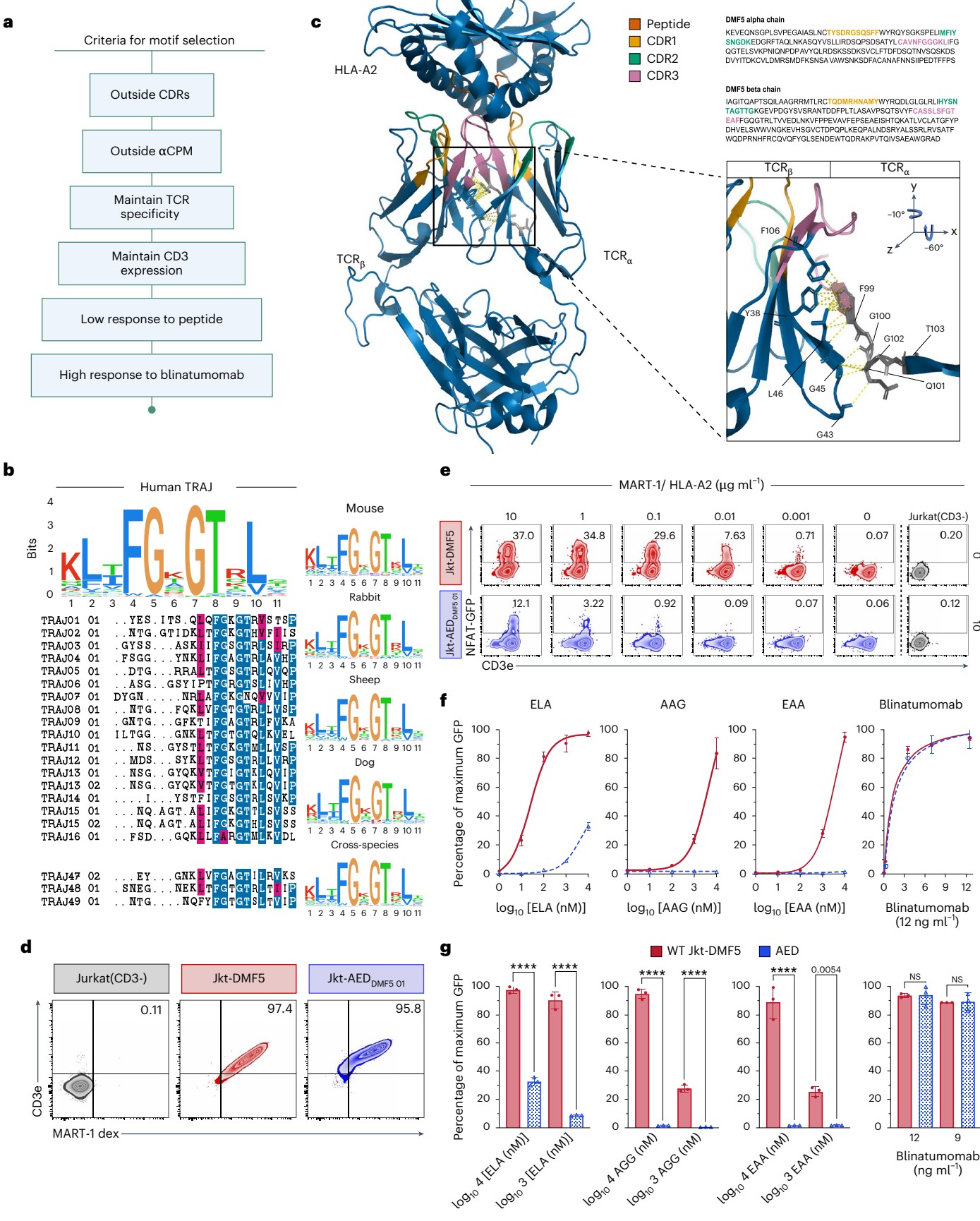

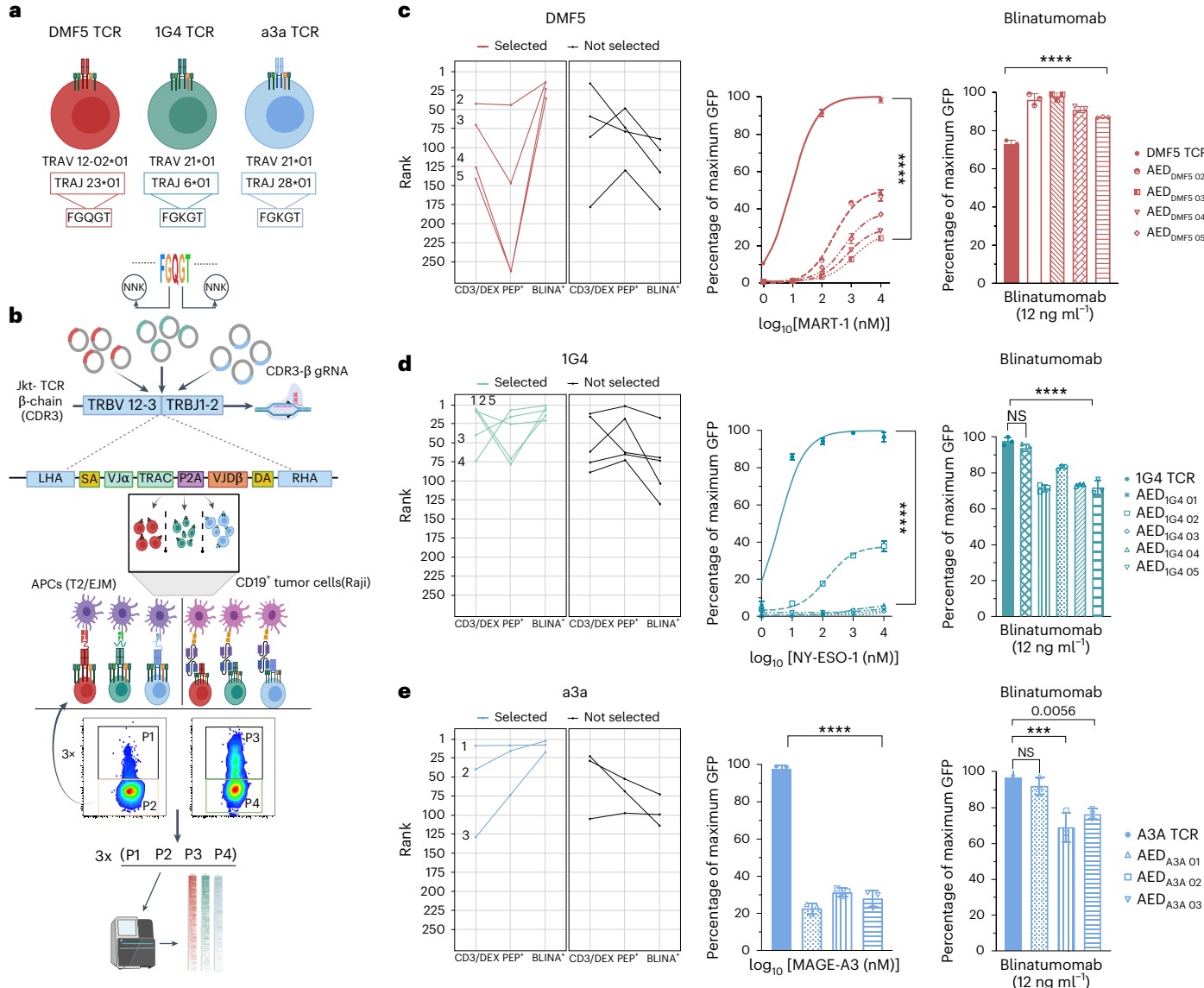

**Fig. 3 | AED T-cell discovery with library mutagenesis and functional screening. a**, Three TCRs with specificity to tumour-associated antigens and their HLA restriction, TCRα V-gene germline (TRAV) and TRAJ usage, and FGxGT sequence motif. **b**, The DMF5 motif (FGQGT) was used for all the libraries. The glycine (G) residues were mutated with NNK mutagenic oligonucleotides yielding a library size of 400 variants for each TCR. First, T cells with maintained CD3 surface expression and pMHC binding were selected by FACS. Next, co-culture assays were performed with either T2 cells pulsed with cognate peptide (DMF5 and 1G4) or with EJM cells (a3a); NFAT-GFP+ CD3+ expressing variants were isolated by FACS. The same selection strategy was applied to cells co-cultured with CD19+ cells and blinatumomab. The fraction with a low response to the peptide (orange square, P2) was pursued in the next round of selection (three

rounds total). Genomic DNA from each group was extracted and submitted for deep sequencing. **c–e**, DMF5 TCR library (**c**), 1G4 TCR library (**d**) and a3a TCR library (**e**). Ranking of variants was followed through DEX/CD3+, PEP+ and BLINA+ populations. Selected variants were primarily selected on their improved ranking in the BLINA+ versus PEP+ population. Selected variants were introduced into Jurkat T cells and individually tested for peptide and blinatumomab response (12 ng ml⁻¹). In **d** and **e**, the experiment was performed once with technical replicates. Data were normalized to the highest GFP signal for each TCR group and are displayed as the mean ± s.d. *P* values were determined using a one-way ANOVA with Tukey's correction for multiple comparisons. *$P < 0.05$; **$P < 0.01$; ***$P < 0.001$; ****$P < 0.0001$; NS, not significant. Significant $P > 0.001$ are numerically written. Panels **a** and **b** were created with BioRender.com.

showcase how the mutations in the FGxGT (TRAJ region) can have a profound effect on the distal components of the TCR/CD3 complex and how AED $_{DMF501}$ mutation attenuates the activation signal coming from the antigen binding (Supplementary Fig. 4a,b).

As the FGxGT motif is present in both the α- and β-chains of TCRs, we set out to determine the effect that the FEQWT mutation would have on the TCR functionality when engineered in the β-chain. The results were unexpected: first, the β-chain variant had a drastically reduced binding to peptide-dextramer (Supplementary Fig. 3e) compared with the WT and Jkt-AED $_{DMF5 01}$. Second, FEQWT-β, regardless of low peptide-binding, had a strong GFP expression and T-cell activation

(Supplementary Fig. 3f), and even more surprising, the response to biAb did not correlate with the response to peptide. We observed a 50% reduction in the GFP expression compared with the WT DMF5 TCR (Supplementary Fig. 3g). Together, these findings showcase the complexity of TCR engineering and the myriad of T-cell functionalities it can create.

## Identification of AED T cells across different TCRs by mutagenesis and functional screening

Structural studies of TCRs have shown substantial similarities in the spatial conformation of TCR α- and β- chains[30,31]. Hence, we initially

introduced the same mutations from the AED$_{DMF501}$ T cells into the α-chain of two additional TCR clones: TCR 1G4, with specificity to tumour-associated antigen NY-ESO-1 (peptide, SLLMWITQC (SLL))[32] and a3a TCR, a previously engineered TCR with high affinity to the melanoma-associated MAGE-A3 antigen (peptide, EVDPIGHLY (EVD))[28,29] (Supplementary Fig. 4a,b). Both TCRs have been investigated in clinical trials of TCR–T-cell therapies; the a3a TCR was discovered to cause a lethal cross-reactivity reaction to an off-target peptide antigen derived from the titin protein expressed on cardiomyocytes[25]. Thus, we aimed to reverse engineer the a3a TCR to be unresponsive to the EVD peptide. DMF5, 1G4 and a3a TCRs use different V-gene and J-gene germlines in the α- and β-chains. In addition, within the FGxGT motif, the 'x' position varies: DMF5 uses glutamine (Q), while 1G4 and a3a use lysine (K), which is the most common amino acid in this position across human J-genes (Fig. 3a and Supplementary Fig. 5a,c). We found that the FEQWT mutation in the 1G4 and a3a motifs either failed to express entirely or showed a substantial loss in surface expression and blinatumomab activation, showcasing that the FEQWT mutation is not compatible for all TCRs. Next, we generated mutagenesis libraries of the FGxGT motif on the backbone of DMF5, 1G4 and a3a. As the WT FGxGT is a highly peptide-responsive variant and due to the inherent abundance of the starting sequence in the library generation, to reduce the difficulty of 'cleaning' the peptide non-responsive population of the WT clone, each TCR library was generated on the FEQWT variant which previously showed decoupled properties (DMF5) or difficulties to express (1G4 and a3a TCRs). Libraries were designed with both G amino acids being replaced with degenerate codons (NNK), resulting in a theoretical diversity of 400 variants per TCR. The TCR libraries containing 400 variants each were integrated into Jurkat cells via CRISPR–Cas9 HDR, as previously described (Supplementary Fig. 1a). Variants were first selected based on both CD3 surface expression and cognate peptide-HLA dextramer binding (CD3/DEX$^+$). Second, variants were functionally screened by co-culture with either APCs expressing cognate peptides or with Raji cells and blinatumomab (Fig. 3b). For DMF5 and 1G4 TCR libraries, T2 cells were pulsed with ELA and SLL peptides, respectively, which are presented on HLA-A*02:01. For the a3a library screening, EJM cells (IgG lambda myeloma plasma cell line) were used as they naturally express the MAGE-A3 antigen and the corresponding EVD peptide on HLA-A*01:01 (Supplementary Fig. 5c). Enrichment and selections were performed by fluorescence-activated cell sorting (FACS) based on the activation-induced GFP expression, and GFP-high and GFP-low populations were sorted (Supplementary Fig. 5d). Subsequently, genomic DNA was extracted followed by targeted PCR amplification of the FGxGT motif (TCR α-chain) and deep sequencing (Illumina MiSeq).

Computational analysis (Supplementary Fig. 6a–c) revealed TCR clones with unique properties and, among them, variants with decoupled peptide-HLA and CD3 activation (Fig. 3c–e and Supplementary Fig. 6d–h). Functional screening requires complex clone evaluation, as TCR activation is not a binary response but rather a continuum. Even in the monoclonal population, the highest activation response does not reach 100% (depending on the TCR and peptide, it can vary from ~20% to ~60%). We followed the dynamic movement of variants in pre-determined subpopulations (P1 to P4), with emphasis on the difference in the rank between peptide (PEP$^+$, P1) and blinatumomab (BLINA$^+$, P3). In the evaluation, it was essential to consider the starting point of the variants, as highly ranked variants in the initial CD3/DEX$^+$ sort would have less room for rank improvement. To further evaluate our clonal selection, several AED-T-cell candidates were independently challenged with antigen (Fig. 3c–e) or Raji cells and blinatumomab (Fig. 3c–e). All selected variants across TCRs showed a substantial reduction in response to cognate peptide antigen and a varying activation to blinatumomab independent from their peptide response, as expected from our previous findings. Selected AED-DMF5 variants exceeded the response of WT to blinatumomab (Fig. 3c). AED-1G4

variants showed a stark reduction in response to cognate SLL peptide (Fig. 3d), and one variant showed equal activation to blinatumomab as the WT 1G4 TCR. It is worth noting that AED variants of the high-affinity a3a TCR showed an 80% reduction in response to the MAGE-A3 peptide (Fig. 3e), and one AED-a3a variant also maintained a WT activation profile in response to blinatumomab. Together these findings show that with functional screening it is possible to derive variants of the FGxGT motif on different parental TCRs and with new activation/signalling properties. Furthermore, unique functional variants can be generated outside of CDRs, where most of the efforts in the TCR engineering field are currently concentrated[31,33].

## Primary human AED T cells show complete activation to blinatumomab with substantially reduced proliferation, cytokine release and cytolysis to antigen stimulation

To further validate our results, we introduced AED$_{DMF501}$ (DMF5-FEQWT variant) into primary human T cells by Cas9-mediated HDR integration. The AED$_{DMF501}$ TCR gene cassette was introduced upstream (5′) of the TCR α-chain constant region (TRAC)[34,35] (Fig. 4a and Supplementary Fig. 7a). Next, AED$_{DMF501}$ T cells and WT DMF5 T cells were isolated by FACS based on double-positive binding to MART-1 dextramer and CD3 surface expression (Fig. 4a). Cells were sorted, and a pure population of cells was obtained (Fig. 4a and Supplementary Fig. 7e). Initially, a smaller fraction of MART-1 dextramer and CD3 double-positive cells in AED$_{DMF501}$ T cells compared with WT DMF5 (Supplementary Fig. 7b) was observed, and overall efficiency was low (~5%). Sequencing results of each subpopulation revealed that mispairing with the endogenous β-chain may be interfering with expression. This phenomenon was even more evident in the AED$_{DMF501}$ cells (Supplementary Fig. 7c). The FGxGT motif is at the TCR αβ-chain interface, and amino acid substitutions required for decoupling of the TCR–antigen binding and CD3 signalling also leads to less preferential pairing with the cognate DMF5 β-chain. Similar results were seen in multiple T-cell donors independent of HLA configuration. Thus, we set out to simultaneously knock out the endogenous β-chain (TRBC locus) while introducing the TCR payload in the TRAC locus. A substantial improvement in surface expression and equivalent expression of the WT DMF5 and AED$_{DMF501}$ was observed (Supplementary Fig. 7b,d). Next, we examined the CD8 to CD4 ratio in the T-cell populations across three donors and discovered that AED$_{DMF501}$ T cells and WT DMF5 T cells had a higher CD8 to CD4 ratio (2:1) compared with WT donor cells (1:2) (Supplementary Fig. 8a,b). This is unsurprising as the DMF5 TCR is naturally associated with the CD8 molecule, and thus both TCRs (DMF5 WT and AED$_{DMF501}$) were more favourably expressed in CD8 T cells. Next, we evaluated in vitro primary T-cell proliferation of AED$_{DMF501}$ T cells and WT DMF5 T cells by performing co-cultures with APC (T2) pulsed with MART-1 peptide antigens (ELA, AAG and EAA). In addition to multiple peptide concentrations, T cells were co-cultured with T2-peptide pulsed cells at a 1:10 ratio (Fig. 4a) to better mimic the abundance (avidity) of antigens in healthy tissue. The experiments were conducted over 5 days, and the T cell to T2 ratio was measured via flow cytometry (Fig. 4b and Supplementary Fig. 8c,d). AED$_{DMF501}$ T cells showed minimal proliferation at the highest concentration of peptide (~5%). By contrast, DMF5 T cells proliferated rapidly and completely overtook the T2 population, reaching more than 95% of the entire cell culture population (Supplementary Fig. 8c). This was due to a specific antigen-driven response, as WT DMF5 T cells did not proliferate to non-peptide presenting T2 cells (Supplementary Fig. 8c,d). Furthermore, we investigated T-cell killing of peptide-pulsed T2 cells for two naturally occurring peptides (EAA and AAG) at a 1:1 ratio with T cells. As previously reported, the physiological expression level of a given peptide-HLA on the cell surface is 10–150 molecules[36], which correlates to a concentration of 0.1 to 10 ng ml$^{-1}$ of peptide pulse. At this level of EAA peptide expression, AED T cells do not induce any appreciable cell killing, whereas DMF5 led to substantial T2 cell death. Although T2 cell death was detected at higher concentrations of AAG,

it should be noted that AAG peptide is predominantly expressed on melanoma tumour cells[37] (Supplementary Fig. 8e).

To examine the cytokine release, we evaluated the secretion of IL-2 (ref. 38), a key regulator of T-cell function and proliferation, as well as interferon-gamma (IFN-γ)[33,39], an essential molecule for cytotoxic activity of CD8 T cells. Assays were performed with three cognate peptides (ELA, AAG and EAA) and T cells derived from three unrelated healthy donors. Enzyme-linked immunosorbent assays (ELISA) revealed IFN-γ secretion levels from AED T cells were significantly lower than WT DMF5 T cells (up to 85%), across peptides (Fig. 4c). In addition, very low levels of IL-2 were produced by T cells across all peptides and their various concentrations. Even at the highest peptide concentration (10 μg ml$^{-1}$), AED$_{DMF501}$ T cells produced ~80% less IL-2 relative to WT DMF5 T cells (Fig. 4d). Next, we examined the lysosomal-associated membrane protein-1 (LAMP-1 or CD107a) expression on WT DMF5 T cells when stimulated with the antigen. Across peptide concentrations, we observed a >50% expression reduction in AED$_{DMF501}$ (Fig. 4e) compared with the DMF5 TCR. In addition, we measured Granzyme B release, a caspase-like protease that is released from T-cell granules upon activation. In the supernatant we observed up to 70% less Granzyme B in AED$_{DMF501}$ cultures compared with WT DMF5 (Fig. 4f). Collectively, these data show that AED$_{DMF501}$ T cells have a strikingly different and safer response profile to cognate peptides compared with the WT DMF5 T cells.

Next, we assessed their capacity of being activated through the CD3 receptor. Thus, we first evaluated T-cell proliferation in co-culture with the CD19$^+$ tumour cell line (Raji B cells) in the presence of blinatumomab (Fig. 4g and Supplementary Fig. 10b). Across all donors and T-cell samples (WT DMF5, AED$_{DMF501}$ and non-engineered donor T cells), we observed no significant difference in T-cell proliferation. A similar trend was observed for IFN-γ secretion (Fig. 4h); however, likely due to the different CD4:CD8 T-cell ratio (Supplementary Fig. 8b,c), donor T cells produced more IL-2 than both AED$_{DMF501}$ and WT DMF5 T cells, but no significant difference was observed between AED$_{DMF501}$ and WT DMF5 (Fig. 4i). In addition, we investigated the CD107a expression and Granzyme B secretion in response to blinatumomab. Contrary to the response to the cognate peptide, AED$_{DMF501}$ T cells had an equal expression of CD107a (Fig. 4j) and secretion of Granzyme B (Fig. 4k) when activated with blinatumomab just like the WT DMF5 TCR and non-engineered donor T cells. Next, we measured tumour (Raji) cell killing in the presence of blinatumomab for two effector to target ratios (E:T) (1:10 and 1:1). The results show that AED$_{DMF501}$ T cells are effective in eradicating tumour cells at 1:1 ratio, similar to the control T cells (DMF5 or non-engineered donor) (Supplementary Fig. 8f). Furthermore, the data clearly show that the *E* to *T* ratio is a determining factor for achieving high cytolysis of tumour cells, as the 1:10 ratio was not sufficient to induce substantial tumour cell killing. Last, we used fluorescence microscopy to capture the interactions between T cells and tumour cells with and without the addition of blinatumomab (Fig. 4l). In all the

groups without blinatumomab, T cells were dispersed around tumour cells and did not inhibit tumour growth (red cluster). Similarly, across all groups, we observe that the treatment with blinatumomab activates T cells leading to their proliferation and enabling them to infiltrate and encircle the tumour cell cluster, thus limiting their growth. We measured the radial intensity difference of these tumour/T-cell clusters across T-cell samples (Fig. 4m and Supplementary Fig. 9a,b). We observed there was no significant difference in the cluster formation between DMF5 WT and AED$_{DMF501}$ T cells or between AED$_{DMF501}$ and non-engineered donor T cells (Fig. 4n and Supplementary Fig. 9c). These data show that despite a disrupted response to peptide stimulation, AED$_{DMF501}$ T cells remain fully functional when engaged with a CD3$^-$ activating molecule such as blinatumomab.

## Primary human AED T cells combined with blinatumomab in xenograft mouse models show potent antitumour immunity and the absence of alloreactivity

Next, we aimed to determine the activity of AED T cells in vivo. We hypothesized AED T cells would be able to clear tumour cells as effectively as conventional T cells (WT donor T cells) when activated with blinatumomab. To this end, we used an established human tumour xenograft mouse model, where immunodeficient nod-scid-gamma mice were engrafted subcutaneously with luciferase (LUC)-expressing Raji (CD19$^+$) tumour cells (Raji–RFP–LUC) and either non-engineered donor T cells or donor-derived AED$_{DMF501}$ T cells. To show the possibility of donor mixing, as AED$_{DMF501}$ T cells do not require MHC matching, each administered product was a mix of two unrelated donors (1:1) (Supplementary Fig. 10a). After tumour/T-cell engraftment, mice received intravenous injections (tail vein) of blinatumomab for five consecutive days; control groups with no blinatumomab administration were also included[40–42] (Fig. 5a). On day 0, each group of mice had a comparable LUC activity (Fig. 5b) and thus comparable tumour engraftment. The experiment was terminated when all the mice in the control group (tumour cells only) reached a terminal bioluminescence signal (>5 × 10$^9$ photons per s). After 7 days post engraftment and 2 days after the last blinatumomab dose, all mice receiving blinatumomab treatment showed no sign of tumour progression and no detectable LUC activity (Fig. 5c,e). As hypothesized, AED$_{DMF501}$ T cells were able to effectively target and clear tumour cells similar to non-engineered donor T cells.

However, unexpectedly, mice receiving the non-engineered donor T cells in the absence of targeted blinatumomab activation cleared the Raji tumours just like the mouse groups receiving blinatumomab (Fig. 5c,e). To discover the reason behind this, on the 28th day, 4 mice from each non-engineered donor group (with and without blinatumomab) were euthanized. However, at this time point, there was no detectable presence of tumour cells or T cells (Supplementary Fig. 10c). Given the fact that Raji cells were not engineered to downregulate HLA-I expression, the most likely explanation for the observed tumour

---

**Fig. 4 | In vitro functional assays of primary human AED T cells. a**, Primary human T cells (AED$_{DMF501}$ and WT$_{DMF5}$) were transfected with the dsDNA/CRISPR–Cas9 and isolated by FACS based on binding to MART-1 dextramer (DEX$^+$) and CD3 expression (CD3$^+$). Sorted T cells were co-cultured with T2 cells pulsed with a range of cognate peptides (ELA, AAG, EAA) at a 1:10 ratio (T to T2 cells). **b**, The proliferation curves for AED$_{DMF501}$ and WT$_{DMF5}$ are plotted for each MART-1 peptide and three different HLA donors. **c,d,f**, Co-culture supernatants were collected and values of IFN-γ (**c**), IL-2 (**d**) and Granzyme B (ELA) (**f**) were measured by ELISA, and dose–response curves are displayed. **e**, Degranulation was measured by CD107a staining upon incubating T cells for 4 h with ELA peptide-pulsed T2 cells (1:1 cell ratio). In **b–f**, symbols are means of three biological replicates. Error bars, s.d. **g–i,k**, T-cell proliferation (**g**), IFN-γ (**h**), IL-2 (**i**) and Granzyme B secretion (**k**) are shown for WT donor T cells, WT$_{DMF5}$ and AED$_{DMF501}$ T cells following co-culture with CD19$^+$ tumour cells (Raji) and blinatumomab. **j**, Degranulation was measured by CD107a staining upon incubating T cells for 4 h with blinatumomab (12 ng ml$^{-1}$)

and Raji cells (1:1 cell ratio). In **g–k**, symbols are means of three biological replicates. Error bars, s.d. **l**, Raji cells were labelled with a red fluorescent dye and were added to the wells containing T cells at a 1:5 ratio. Microscopy images depict the Raji cell cluster size difference between +blinatumomab and −blinatumomab samples, and a formation of T-cell rings is observed around Raji cells when blinatumomab is added. **m**, The radial intensity difference shows T-cell activation around Raji clusters with (solid line, dark) and without (dashed line, light) blinatumomab (*n* = 12); s.d. (shaded area) **n**, The intensity difference between Raji cells and T cells is shown for donor, DMF5 and AED$_{DMF501}$ cultured with and without blinatumomab. In the box plot, the line represents the median and whiskers minimum and maximum values measured. Dots represent 12 clusters measured across three technical replicates (one donor). *P* values were determined with one-way ANOVA with Tukey's correction for multiple comparisons. *$P < 0.05$; **$P < 0.01$; ***$P < 0.001$; ****$P < 0.0001$; NS, not significant. Significant $P > 0.001$ are numerically written. Panel **a** was created with Biorender.com.

rejection in mice that did not receive blinatumomab is that there was an alloreactive T-cell response against the MHC-mismatched Raji-tumour cell line. On the contrary, $AED_{DMF501}$ T cells without co-administration of blinatumomab did not induce an antitumour cell response (sustained growth of Raji cells in 4 out of 5 mice). These in vivo data suggest that $AED_{DMF501}$ T cells do not drive an alloreactive response (Fig. 5d,e) and remain idle without the presence of a CD3-activating molecule such as blinatumomab. Mice receiving $AED_{DMF501}$ and blinatumomab were

tumour-free for the duration of the experiment (42 days). Mice weight was consistently monitored, and mice in all of the mouse groups did not show signs of xeno-alloreaction (Supplementary Fig. 10d). Mice that did not receive blinatumomab gradually reached the endpoint of tumour growth (except for the group receiving non-engineered donor T cells). Detectable tumour tissue from mice not receiving blinatumomab was extracted and submitted to immunofluorescence and chromogenic staining (Fig. 5g and Supplementary Fig. 10c). The tissue slides revealed

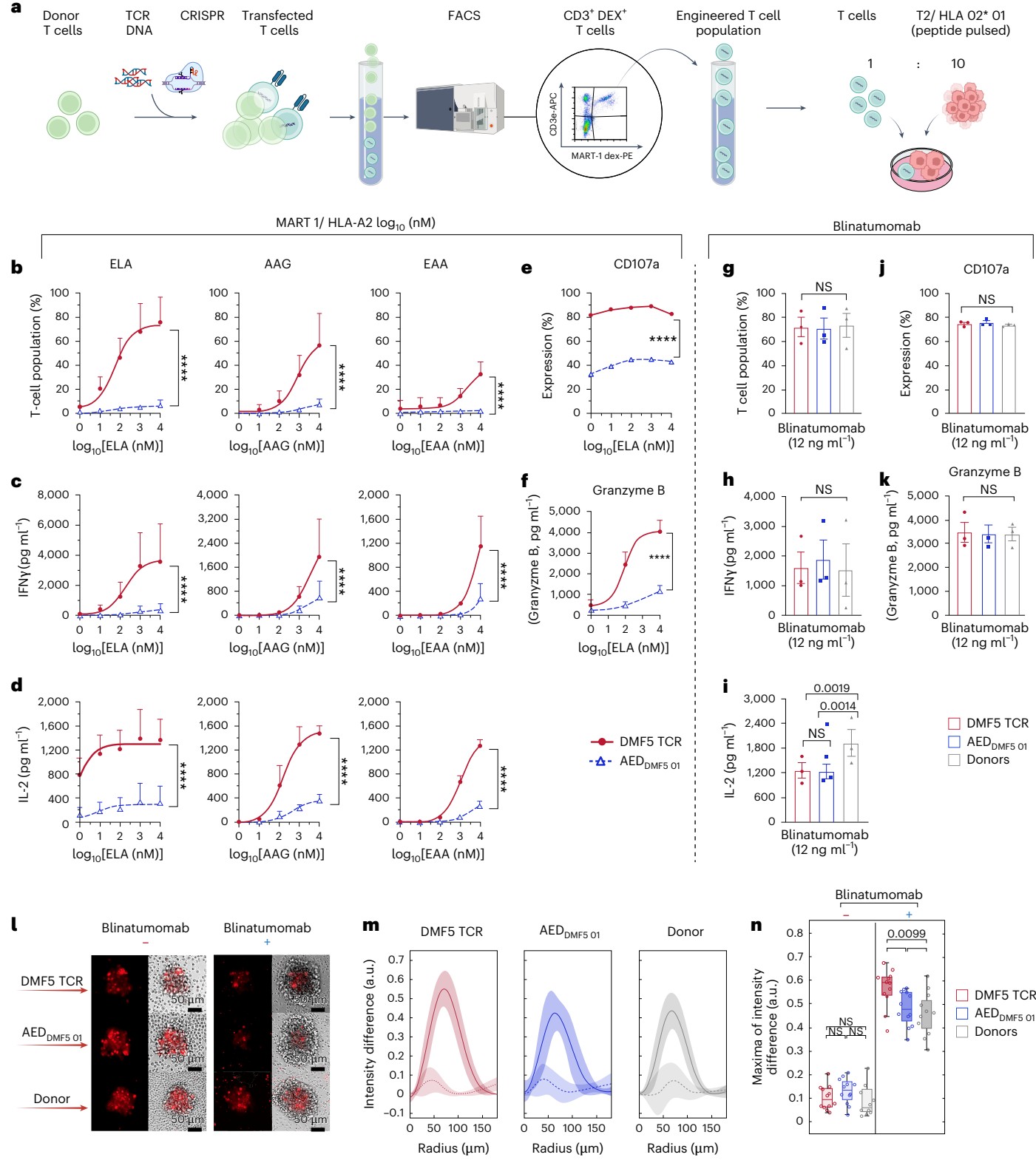

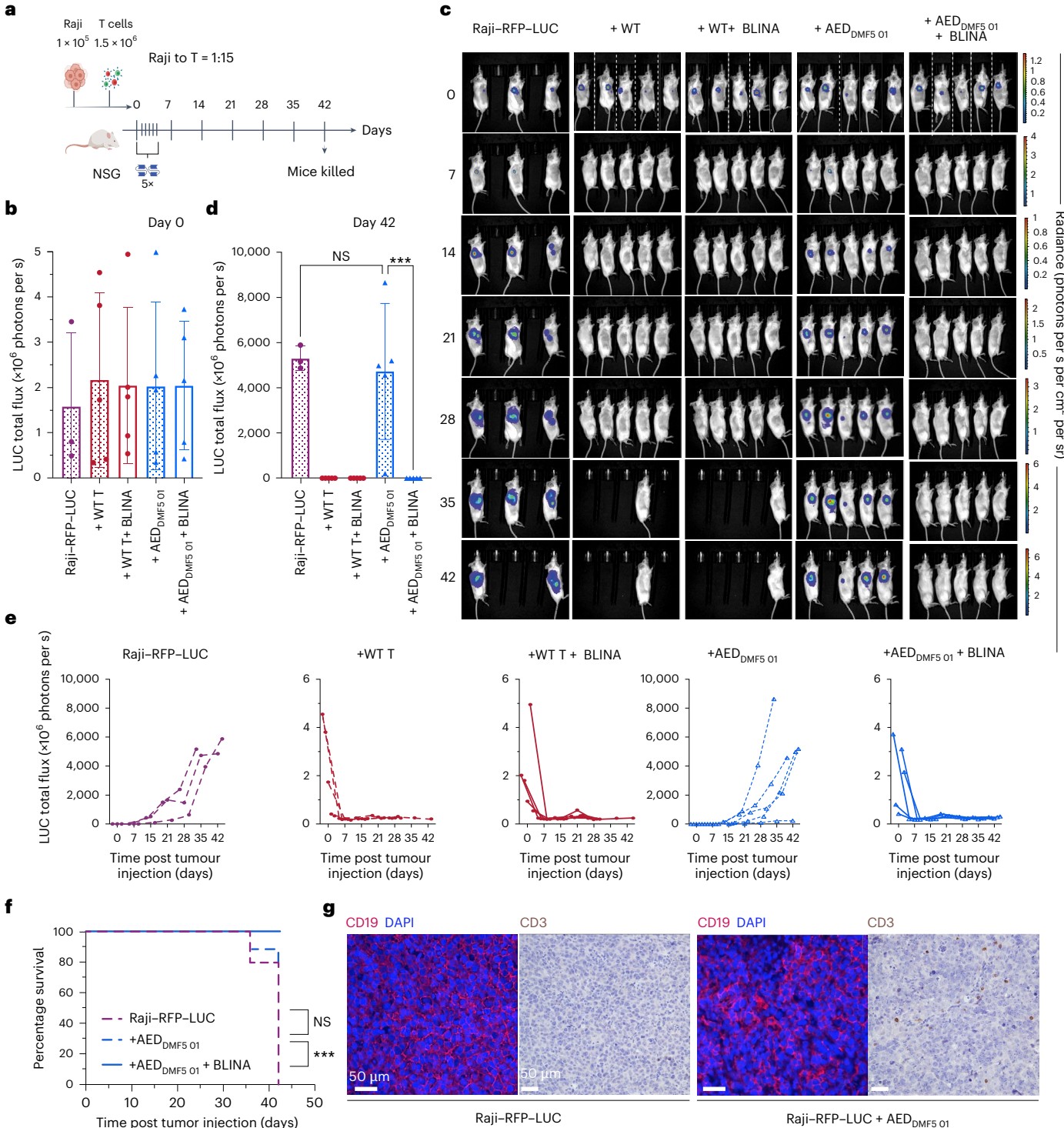

**Fig. 5 | AED T cells with blinatumomab drive potent antitumour response and do not show alloreactivity. a**, Schematic of the human tumour xenograft mouse model. All cell injections were performed subcutaneously with a 1:15 Raji to T-cell mix. WT T cells were a 1:1 mix of two donors (UDN001 and UDN002). Blinatumomab (0.1 µg) was administered intravenously once a day for five consecutive days. Mice were killed on day 42 when the control group (Raji only) reached the highest allowed level of LUC (5 × 10⁹ photons per s). NSG, nod-scid-gamma. **b**, Bioluminescence activity in each mouse group was measured on day 0 by in vivo imaging system (IVIS) imaging. **c**, Tumour progression was followed with weekly IVIS imaging at specified time points. Four mice in each group with WT T cells were killed on day 28 to determine the complete loss of bioluminescence. **d**, Tumour bioluminescence was measured at the end of the experiment (day 42). Mice were killed at an earlier time point, and their values are also plotted. **e**, Tumour bioluminescence signals are measured at specified time points. Individual lines denote data obtained from each animal. In **a–e**, the experiment was performed once with two donors. Symbols are means of three to five mice. Error bars, s.d. **f**, Kaplan–Meier curves show the overall survival of mice in the selected experimental groups. *P* values were determined using a two-sided log-rank test. **g**, Immunohistochemistry (CD19) and chromogenic staining (CD3) of representative mice in the mouse groups with tumour cell overgrowth. In **d**, *P* values were calculated using one-way ANOVA with Sidak's correction for multiple comparisons. *\*P* < 0.05; \*\**P* < 0.01; \*\*\**P* < 0.001; \*\*\*\**P* < 0.0001; NS, not significant. Significant *P* > 0.001 are numerically written. Panel **a** was created with Biorender.com.

a sparse presence of AED$_{DMF501}$ T cells within a large majority of tumour cells. Characteristic T-cell clusters indicative of T-cell activation and proliferation were not observed. Together, these findings support the safety potential of AED$_{DMF501}$ T cells in adoptive T-cell therapy.

## Discussion

We have provided evidence that it is possible to functionally decouple TCR–antigen binding from the CD3 signalling axis and generate human AED T cells. AED T cells showed remarkable unresponsiveness to cognate antigen stimulation while being fully activated when a biAb engages their CD3 receptors. Furthermore, AED T cells, in the presence of blinatumomab, effectively cleared Raji tumour cells in vivo and showed no signs of alloreactivity. These results support a strategy for the development of off-the-shelf donor T cells that can be effectively and safely combined with biAb therapies.

The folding and assembly of the TCR–CD3 complex and its associated components is a tightly controlled mechanism with multiple checkpoints[17,22]. The functional sequence landscape outside of the CDR regions and constant domains has not been extensively explored in previous studies, although the FGxGT motif has been previously identified as conserved and speculated to have a role in TCR–pMHC engagement[18,19]. In a comprehensive germline sequence analysis of the human TCR V- and J-gene sequences, we confirmed that the FGxGT motif in the TCR J-genes is highly conserved and, through functional assays, has a critical role in TCR signal transduction. Although future structural studies would add value for elucidating the mechanistic effect of AED mutations, our functional findings and initial molecular dynamics simulation show that it is possible to introduce mutations outside the CDRs while maintaining the structural integrity of the TCR/CD3 complex, preserving TCR–pMHC specificity and yet drastically changing the functional and activation properties of TCRs. We hypothesize that the structural changes caused by mutations in the glycine residues of the FGxGT motif result in mechanically unfavourable configurations for the proper signal transduction to CD3 molecules. As a very similar motif is also found to be highly conserved in antibody variable heavy (WGxGT) and light chains (FGxGT)[43], it is plausible that mutational variants and structural changes may have an impact on B cells as well (as in antibody and B-cell effector functions). In addition to FGxGT, we also identified other candidate motifs that were highly conserved in V-gene germlines of the TCR α- and β-chains (WYxQ), which could also be explored for their functional impact on TCR–antigen binding and CD3 signalling.

To screen and identify mutational variants capable of decoupling the TCR–CD3 complex, we designed mutagenesis libraries in the FGxGT motif for three clinically investigated TCRs (DMF5, 1G4 and a3a). With functional assays, in each library, we discovered a panel of variants with a diverse profile of TCR–antigen and CD3 activation. Novel variants with their distinctive properties may provide advantages in specific clinical settings either with other biAbs (biAbs with different affinities or binding epitopes of CD3) or as unique T-cell products for applications outside cancer immunotherapy (autoimmune disease). For T-cell activation, both TCR affinity and avidity have an effect. For example, tumour-associated antigens are over-expressed on tumour cells relative to healthy cells. Thus, a high-affinity TCR, such as DMF5 TCR, has been clinically investigated as a TCR–T-cell therapy against highly expressing MART-1$^+$ melanomas[25]; however, the high-affinity DMF5 TCR led to unwanted on-target/off-tumour toxicities in the healthy tissue and adverse safety effects (skin rash, uveitis and hearing impairment). To this end, we used a wide range of peptide concentrations ($10^{-5}$ M to $10^{-9}$ M) to better understand the activation threshold of AED$_{DMF501}$ T cells. In previous studies[36], it was determined that a natural expression of peptide antigen can be reproduced with T2 cells pulsed with $10^{-8}$ M to $10^{-10}$ M (0.1 o 10 ng ml$^{-1}$) peptide concentration. In this peptide range, AED$_{DMF501}$ T cells especially with the EAA peptide were not proliferating or killing the peptide-pulsed T2 cells, highly suggesting AED$_{DMF501}$ T cells

could be safe from GvHD. However, further pre-clinical development, safety testing and dosage optimization would be necessary to confirm the acceptable safety profile of the AED$_{DMF501}$ variant. Our in vivo study showed that AED$_{DMF501}$ T cells do not cause an alloreactive response such as the one observed in non-engineered donor T cells, yet AED$_{DMF501}$ T cells were still fully capable of targeting and clearing tumour cells when in the presence of a blinatumomab.

In its current form, we envision AED T cells in combination with biAbs as being beneficial for highly immunocompromised patients (for example, early-relapse patients post haematopoietic stem cell transplantation), particularly in the first 100 days, who often lack a sufficient number of functional T cells due to the transplantation pre-conditioning and previous rounds of chemotherapy[44]. However, for such a therapy to be effective in immunocompetent patients, host-versus-graft disease (HvGD) could be a considerable problem that needs to be addressed. HvGD is not only a challenge for AED T cells but also an inherent risk for other allogeneic cell therapies that are under clinical investigation, including CAR T cells and TCR-T cells[45]. Several strategies have been described to mitigate host rejection. Recently, an alloimmune defense receptor (ADR)[46] was engineered to induce the deletion of activated host T and natural killer cells. The ADR receptor was combined with CARs to generate an allogeneic T-cell therapy with promising preclinical results. ADRs and other cellular engineering approaches (such as HLA-I KO, HLA-E, CD47 overexpression, CD52 KO)[47–49] could be multiplexed with AED T cells to simultaneously address GvHD and HvGD challenges.

At the same time, biAbs are rapidly evolving, including the engineering of more complex multispecifics and combinations of biAbs that, in addition to targeting CD3, also target other co-stimulatory receptors on T cells (4-1BB, CD28; ref. 50). This strategy addresses the need for co-stimulatory signals to achieve sustained and improved T-cell engagement, an important quality of CAR- T-cell therapies. However, CAR-T cells run into the problem of persistent antigen stimulation much like the endogenous T cells, which can drive T cells into exhaustion[51].

A combination approach of healthy AED T cells with improved bispecific engagers could have a substantial clinical impact as AED T cells could be, in principle, stimulated multiple times and even easily switch targets as the tumour antigen landscape changes. Furthermore, T cells can rest in between stimulations, which would likely increase their longevity and the duration of the therapeutic effect.

CAR-T-cell technologies have had substantial clinical success; however, more treatment options across different clinical indications and patient groups is still an unmet medical need. Most modalities of immunotherapy (CAR-T, TCR-T, biAb and immune checkpoint inhibitors) are dependent on the number and function of patient T cells. Preclinical studies with the biAb mosunetuzumab (anti-CD20 × anti-CD3) showed ~45% of patients did not have a sufficient number of functional T cells, thus leading to a lack of therapeutic efficacy[7]. The efficacy of mosunetuzumab could only be restored with the addition of healthy donor T cells. Currently there are eight clinically approved biAbs and another hundred plus molecules with different targets and biAb formats in preclinical and clinical development, nearly all of which depend on the CD3-based activation of T cells[52,53]. As AED T cells have showed intact CD3 signalling across different T-cell-engager formats, they have the potential to be combined with and to enhance a myriad of CD3-targeting biAbs, including a recently approved TCR–CD3 bispecific-engager therapy (Kimmtrack, Immunocore)[54].

## Methods
### Constructs
The sequence of all wild-type TCR clones (DMF5, IG4 and a3a) was ordered as gene fragments (Twist Bioscience). Briefly, each HDR template consisted of homology arms, a P2A sequence, signal peptide and a complete αβTCR separated with a T2A sequence which was cloned

into a pUC19 backbone plasmid (Addgene, 50005) via Gibson assembly (NEB, E2611). Individual AED constructs from libraries were generated with site-directed mutagenesis. These plasmids were used for HDR template amplification (Kapa Hotstart polymerase). Double-stranded DNA HDR templates for transfection were column-purified with DNA clean and concentration kit (Zymo Research, D4013), and the concentration was determined by NanoDrop 2000c spectrophotometer (Thermo Fisher, ND-2000) and concentrated to ~1 µg µl⁻¹ by vacuum concentrator (Eppendorf, 5305000703).

## Cell lines

The Jurkat leukaemia E6-1 T-cell line was obtained from the American Type Culture Collection (ATCC) (TIB152). Jurkats were genomically modified into a TnT TCR display platform (Cas9⁺, CD8⁺, NFAT-GFP⁺, FAS-L⁻, CD3⁻ and CD4⁻)[24] before AED experiments; the T2 hybrid cell line (ACC598) and the EJM multiple myeloma cell line (ACC560) were obtained from the German Collection of Cell Culture and microorganisms (DSMZ), and Raji human Burkitt's Lymphoma cell line was obtained from ATCC (CCL-86); TnT-Jurkat T cells, T2-cells and Raji cells were cultured in ATCC-modified Roswell Park Memorial Institute (RPMI) 1640 (Thermo Fisher, A1049101), and EJM cells were cultured in Iscove's modified Dulbecco's medium (Thermo Fisher, 12440053). All media were supplemented with 10% fetal bovine serum (FBS), 50 U ml⁻¹ penicillin and 50 µg ml⁻¹ streptomycin. All cell lines were cultured at 37 °C, 5% $CO_2$ in a humidified atmosphere and routinely tested for *Mycoplasma* contamination. Cells were passaged every 3 days at a ratio of 1:5 to keep the cell concentration under $1 \times 10^6$ cells per ml. Detachment of EJM adherent cell lines for passaging was performed using the TrypLE reagent (Thermo Fisher, 12605010).

## Comparative modelling and molecular dynamics simulations

The three-dimensional models of the DMF5_wt and AED$_{DMF501}$ TCR−CD3 complexes were generated using the Modeller software (v.10.4)[55]. The protein structure of PDB entry 6JXR served as a template for the comparative modelling. Molecular dynamics simulations were conducted to study the TCR−CD3 complex embedded in an asymmetric lipid bilayer, representing the plasma membrane, following the outlined protocol[56]. The membrane system consisted of two leaflets: the outward-facing leaflet comprised 1-palmitoyl-2-oleoylphosphatidylcholine (POPC, 33.3 mol%), palmitoyl sphingomyelin (PSM, 33.3 mol%) and cholesterol (33.3 mol%), while the cytosolic leaflet comprised POPC (35 mol%), 1-palmitoyl-2-oleoyl-sn-glycero-3-phosphoethanolamine (POPE, 25 mol%), 1-palmitoyl-2-oleoyl-sn-glycero-3-phosphoserine (POPS, 20 mol%) and cholesterol (20 mol%). All molecular dynamics simulations were performed using GROMACS 2023.1 (ref. [57]) with the CHARMM36m force field[58] for solutes, in combination with the TIP3P water model. The simulation systems were prepared using CHARMM-GUI[59,60]. The protein–lipid systems were solvated in a water box, and the lipid bilayer was oriented in the Z plane. Na⁺ and Cl⁻ ions were added to the system at a concentration of 150 mM to neutralize the charge. The simulation systems underwent energy minimization using the steepest descent algorithm until convergence was achieved, with a tolerance of 1,000 kJ (mol nm)⁻¹. The equilibration of the systems was performed in the NVT ensemble (constant number of particles (N), volume (V) and temperature (T)) for 100 ps, maintaining the temperature at 300 K using the $v$-rescale thermostat[61] with a time constant of 1 ps. Following the NVT equilibration, the systems were further equilibrated in the NPT ensemble (constant number of particles (N), pressure (P) and temperature (T)) for 400 ps. The pressure was maintained at 1 bar using semi-isotropic pressure coupling with the Berendsen barostat[62] with a time constant of 1 ps and a compressibility of $K = 4.5 \times 10^{-5}$ bar⁻¹. For the specific purpose of calculating root-mean-square fluctuation values, a simulation duration of 1 ns was deemed sufficient to capture the inherent fluctuations in protein dynamics with the pressure maintained at 1 bar using the Parrinello–Rahman barostat[63] with a time

constant of 5 ps. The analysis of the molecular dynamics trajectories was performed using the Python packages MDAnalysis (2.5.0)[64] and NumPy (1.21.5)[65].

## CRISPR−Cas9 genome editing of cell lines

Transfection of Jurkat-derived cell lines (TnT−Cas9⁺) was performed by electroporation using the 4D-Nucleofector device (Lonza, AAF-1003X) and the SE cell line kit (Lonza, V4XC-1024). Before transfection, single-guide RNA complexes were generated by a 1:1 mix of 2.5 µl of custom Alt-R crRNA targeting Jkt- TRB CDR3 sequence (TCGACCT-GTTCGGCTAACTA) (200 µM, IDT) and 2.5 µl of Alt-R tracrRNA (200 µM, IDT, 1072534) following IDT instructions. For the transfection, cells were maintained at a density between $5 \times 10^5$ and $1 \times 10^6$ cells per ml. About $1 \times 10^6$ cells were washed twice with room-temperature PBS and resuspended in 100 µl of SE buffer together with 1 µg of the HDR template and 5 µl of sgRNA complex. The cell suspension was mixed gently and transferred into a Lonza electroporation cuvette. Cells were electroporated using program CK116 and were immediately topped with 0.5 ml of prewarmed complete media and allowed to rest for 10 min before transferring into a 12-well plate with a Lonza transfer pipette. For Jurkat T cells, Alt-R HDR enhancer (IDT, 1081073) was added at 30 µM final concentration and removed after 14 h by centrifugation. HDR efficiency was assessed by flow cytometry 4 days post transfection.

## Primary human T cells culture and genome editing

Peripheral blood mononuclear cells were isolated from whole blood of healthy human donors (Blutspendezentrum SRK beider Basel, Universitätspital Basel) via Lymphoprep (Stemcell Technologies, 07861), a standard Ficoll-based density gradient centrifugation. Human CD4⁺ and CD8⁺ T cells were extracted by magnetic negative selection using an EasySep Human Pan T Cell Isolation kit (STEMCELL Technologies, 17951). Primary T cells were cultured in XVivo-15 medium (Lonza, BE02-060F) with 5% FBS and 50 µM 2-mercaptoethanol with freshly added 200 IU of recombinant human IL-2 (Peprotech, 200-02), 100 µg ml⁻¹ Normocin (Invivogen, ant-nr-1). Throughout the culture period, T cells were maintained at $1 \times 10^6$ cells per ml of media. Every 2–3 days, additional media and IL-2 were added, and cells were transferred to larger culture vessels as necessary. On the day of thawing and magnetic selection, T cells were activated with anti-CD3/anti-CD28 Dynabeads (Thermo Fisher, 11456D). Before transfection (day 2), beads were magnetically removed. About 5 µl (200 µM) of assembled sgRNA (targeting TRAC locus (ref. [29])-AGAGTCTCTCAGCTGGTACA) and 5 µl (200 µM) of sgRNA targeting TRBC (AGAGATCTCCCACACCCAAA) were mixed with 1 µl of recombinant SpCas9 (61 µM, IDT, 1081059) and incubated for ~10 min at room temperature. Cas9 RNP complex (6 µl) targeting TRAC locus were added to cells ($2 \times 10^6$) resuspended in 100 µl of P3 Primary Cell transfection buffer (Lonza, V4XP-3032) and were transfected using the EO115 electroporation program. About 600 µl of FBS-free XVivo-15 media was added to the Lonza cuvettes, and cells were incubated at 37 °C (30 min). Cells and media were transferred to a 12-well plate and supplemented with IL-2.

## Flow cytometry

Samples were acquired on either LSRFortessa (BD Biosciences) or CytoFLEX (Beckman Coulter) cytometers, and data were analysed using FlowJo v.10 software. The following antibodies were used in this study. From Biolegend: APC-CD3e (clone UCHT1 300458) PE-Cy7- CD3e (clone UCHT1, 300420), APC- CD4 (clone RPA-T4, 300552), PE-CD8a (clone HIT8a, 300908), PE-Cy7-CD19 (clone HIB19, 302216), PE-conjugated anti-human TCR α/β (clone IP26, 306707) and PE anti-human CD107a (LAMP-1) antibody (clone H4A3, 328608). DAPI viability dye (Thermo Fisher, 62248) was added to antibody cocktails at a final concentration of 1 µg ml⁻¹. Cells were washed once in flow cytometry buffer (PBS, 2% FBS, 2 mM EDTA) before staining, stained for 20 min on ice and washed twice in flow cytometry buffer before analysis. In co-culture

experiments before additional staining reagents, Fc receptors on T2 cells were blocked with TruStain FcX reagent (BioLegend, 422301). Staining with pMHC dextramers was performed for 10 min at room temperature, followed by addition of surface staining antibodies and incubation for 20 min on ice. The following pMHC dextramers were commercially obtained from Immudex: NY-ESO-1$_{157-165}$ (SLLM-WITQC, HLA-A*0201, WB2696-PE); MART-1$_{26-35(27L)}$ (ELA, HLA-A*0201, WB2162-PE); and MAGE-A3$_{168-176}$ (EVDPIGHLY, HLA-A*0101, WA3249-PE). pMHC dextramers were used at a 3.2 nM final concentration (1:10 dilution) for staining. Cell sorting (FACS) was performed using BD FACSAria III or BD FACSAria Fusion instruments. All the samples were sorted in bulk and used as such to avoid variations in signal and cell behaviour arising from single-cell variability.

### Peptides and peptide pulse

Peptides were generated by custom peptide synthesis (Genscript), re-suspended at 10 mg ml$^{-1}$ in DMSO and placed at −80 °C for prolonged storage. For peptide pulsing, T2 cells were collected and washed twice in serum-free RPMI 1640 (SF-RPMI). Peptides were diluted to 10 µg ml$^{-1}$ in SF-RPMI (or to concentrations indicated in figure legends), and the solution was used to make 10-fold dilutions in which cells were resuspended at a concentration of $1 \times 10^6$ cells per ml. Cells were incubated for 120 min at 37 °C, 5% $CO_2$, washed twice with SF-RPMI, resuspended in complete media and added to co-culture wells.

### APC and Jurkat T-cell co-culture assays

T2 cells or EJM were used as APCs in co-culture experiments with either Jurkat WT TCRs or AED-modified TCRs. T cells were at ~$1 \times 10^6$ cells per ml when collected, pelleted by centrifugation and re-suspended in fresh complete media at $1 \times 10^6$ cells per ml and counted. If not stated otherwise, $1 \times 10^5$ T cells (100 µl) were seeded in 96-well plate (V bottom) wells. Antigen-expressing cells (EJM) or peptide-pulsed cells (T2) were adjusted to $1 \times 10^6$ cells per ml in complete media, and $5 \times 10^4$ cells (50 µl) were added to each well with an APC to T-cell ratio of 1:2. Anti-human CD28 antibody (clone CD28.2, 302933; BioLegend) was added as a co-stimulatory signal at a final concentration of 1 µg ml$^{-1}$ to all samples (including negative controls). Plates were incubated overnight at 37 °C, 5% $CO_2$. The next day, expression of NFAT-GFP in modified Jurkat T cells was assessed by flow cytometry.

### Co-culture of primary T cells and ELISAs

WT and AED TCR-reconstituted primary T cells were FACS sorted, supplemented with IL-2 and allowed to rest for 3 days before the co-culture experiment. After resting, T cells were washed, counted and resuspended in complete primary T-cell media. T cells and T2 cells were mixed at a 1:10 ratio ($5 \times 10^3$ and $5 \times 10^4$ cells) in a total of 150 µl of media and incubated overnight at 37 °C, 5% $CO_2$. Cells were cultured for 5 days. Afterwards, the supernatant was collected, and cells were assessed by flow cytometry. Concentration of human IL-2 and IFN-γ, and Granzyme B cytokines were quantified using standard kits (Thermo Fisher, 88-7025-88, 88-7316-88, BMS2027-2). Supernatants were diluted in media to fall within the standard curve for the assay. Negative control values were subtracted from each sample point, and the concentration was calculated from the standard curves. Measured concentrations of cytokine were plotted versus the peptide concentration and fitted to a four-parameter logistic model.

### Cytotoxicity assays of primary T cells

On day 7 post transfection, T cells transfected with AED or DMF5 TCRs were FACS sorted and allowed to rest for 1 day before the co-culture experiment. Untransfected T-cell controls, AED or DMF5 TCR transfected T cells were co-cultured with Raji cells in the presence or absence of blinatumomab at 1:10 and 1:1 T cell-to-target cell ratio or with peptide pulsed T2 cells (1:1). After 4 days of co-culture, cells were stained with CD19-APC and CD3-PE antibodies (Biolegend) and analysed by flow

cytometry. Sytox Blue viability dye (Invitrogen, S34857) was used to exclude dead cells from the analysis. CountBright Plus counting beads (Invitrogen, C36995) were used to calculate the absolute numbers of target cells (Raji or peptide-pulsed T2 cells) after 4 days of co-culture.

### Cell stimulation and staining for CD107a expression

To measure degranulation in response to target-cell stimulation, effector cells were incubated with target cells (Raji) at a 1:1 ratio. CD107a (PE) antibody was added to the culture before stimulation. Unstimulated cells were incubated without target cells to detect spontaneous expression of CD107a. Cells were incubated at 37 °C in 5% $CO_2$ and analysed at a 4 h time point. The cells were washed twice with PBS and stained using conjugated antibody, CD3 (APC), for 15 min at room temperature.

### TCR library design, selection and sequencing

Deep mutational scanning combinatorial libraries of the TRAJ motif (FGxGT) libraries for TCR$_{DMF5}$, TCR$_{IG4}$ and TCR$_{a3a}$ were generated by plasmid nicking mutagenesis as previously described[47]. Briefly, the protocol relies on the presence of a single BbvCI restriction site for sequential targeting with Nt.BbvCI and Nb.BbvCI nickases, digestion of wild-type plasmid and plasmid re-synthesis using mutagenic oligonucleotides. Mutagenic oligonucleotides were designed using the QuikChange Primer Design online tool (Agilent). After nicking mutagenesis, mutated plasmids were transformed into 100 µl of chemically competent *Escherichia coli* DH5α cells (NEB, C2987H) and plated on ampicillin (100 µg ml$^{-1}$) LB agar in Nunc BioAssay dishes (Sigma-Aldrich, D4803). Serial dilutions of transformed cells were plated separately to quantify bacterial transformants. Plasmid libraries were purified from bacterial transformants using the ZymoPURE Plasmid miniprep Kit (Zymo Research, D412). HDR templates were amplified from plasmid libraries by PCR and column-purified before transfection.

Deep mutational scanning library HDR templates and CDR3B guide RNA were used to transfect $1 \times 10^6$ lab-modified Jurkat T cells[24]. First, cells were stained with (dextramer) and FACS sorted for the CD3 surface expression. In the second round, cells were challenged separately with their cognate peptide (0.1 µg ml$^{-1}$) EJM cells and blinatumomab (12 ng ml$^{-1}$), and both GFP$^-$ and GFP$^+$ fractions were sorted (SEL1). GFP-peptide fraction was used as the starting population for following selections (SEL 2 and 3). Genomic DNA from all sorted populations was extracted via PureLink Genomic DNA Mini Kit (ThermoFisher, K182002). Regions of interest were PCR amplified with added TruSeq adapters for 300-PE v3 (600 cycles). MiSeq sequencing was performed in the Genomics Facility Basel.

### In vitro microscopy

Raji ($5 \times 10^4$) and T ($1 \times 10^4$) cells were plated in a 96 half-area well plate (Corning, CLS3690) with a transparent glass bottom for higher sensitivity. Cells were plated in the X-Vivo media without phenol red (Lonza, 04-744Q) and supplemented with FBS. The well plate was placed in an environmental chamber, which provided a 5% $CO_2$ atmosphere and a humidity of at least 70%. The imaging of the cells was conducted on a fully automated Nikon Ti2 microscope with a ×10 magnification (Plan Apo λ10x). Every well containing cells was imaged fully by stitching 3 × 3 images together with an overlap of 15%. The cells were imaged at 24 h and 96 h. Raji cells were stained before the start of the experiment with the CellTrackerTM Deep Red (ThermoFisher, C34565) following manufacturer's protocol. Stained Raji cells were visualized using the mCherry filter cube from Nikon with an exposure time of 50 ms and light power of 33%.

### Image analysis of the Raji/T-cell co-culture

From every condition, 12 clusters of Raji cells were selected. The images were smoothed with a Gaussian filter ($\sigma = 14.4$ µm), and the radial intensity profiles of the clusters were calculated. The intensity profiles of the bright-field images and the red-fluorescence images were normalized with the minimum and maximum intensity values and subtracted to

achieve the T-cell density around the Raji cells. For every cluster, the maximum intensity difference is calculated and used for one-way analysis of variance (ANOVA) to determine the *P* value. All image modification and calculations were computed with MATLAB.

## Data analysis and visualizations
Data analysis was performed using R (version 4.0.1.)[66]. Visualizations were generated using the R packages ggplot2 (ref. [67]) (version 3.3.3) and ggseqlogo[68] (version 0.1, Sequence logo plots). TCR structures were prepared using PyMOL[69], and complete figures and graphics were generated using BioRender software.

## MSA
Germline gene sequences for TRAJ, TRAV, TRAC, TRBJ, TRBV and TRBC were obtained for various species from IMGT (international ImMunoGeneTics information system)[70]. MSA (multiple sequence alignment) was performed for each of these regions within and across species using R-package msa[71] (version 1.20.1, method is 'ClustalW') in R (version 4.0.1).

## Sequencing analysis
Raw sequencing data from screening libraries were preprocessed and aligned using the MiXCR (v3.0.12)[72] software package. Data were cleaned to only contain sequences that showed variation in the positions targeted for mutation. Frequency and rank of unique variants was calculated from clone count. Sequences of interest were identified by a decrease in rank and de-enrichment (based on frequency) in the peptide positive fraction and maintenance of rank and absence of de-enrichment in the blinatumomab positive population.

## Mouse strains and study approval
NOD/SCID/IL-2Rγ-null mice were purchased from Charles River Laboratories. Mice were maintained and bred in the EPFL (École Polytechnique Fédérale de Lausanne) animal facilities in a pathogen-free environment. All animal experimentations were performed in accordance with the Swiss Federal Veterinary Office guidelines and as authorized by the Cantonal Veterinary Office (animal license). Both female and male littermates (aged 5 weeks) were used in the experiments.

## In vivo mouse CD19$^+$ xenograft tumour model
Mice were inoculated in the flank with a mixture of $1 \times 10^5$ Raji–RFP–LUC cells and $1.5 \times 10^6$ WT or AED$_{DMF501}$ T cells. Control group (without T cells) received only $1 \times 10^5$ Raji–RFP–LUC cells. Before the inoculation, cells were washed and resuspended in 100 μl PBS. Before blinatumomab treatment, mice were divided into groups of 5 mice each (Control group had 3 mice), with equal tumour size distribution based on bioluminescent imaging. Blinatumomab (0.1 mg per mouse per injection) was administered through tail-vein injection every day over the course of 5 days following tumour engraftment. Mice's health and weight were monitored three times per week using body and health performance score sheets.

## Bioluminescence imaging
Tumour growth was monitored by bioluminescent imaging. Bioluminescent imaging was performed using the Xenogen IVIS Lumina II imaging system. Briefly, mice were injected i.p. with D-luciferin (150 mg per kg stock, 100 μl of D-luciferin per 10 g of mouse body weight) resuspended in PBS and imaged under isoflurane anaesthesia after 5–10 min. A pseudocolour image representing light intensity (blue, least intense; red, most intense) was generated using Living Image v.4.5 software (Caliper Life Sciences). Mice were killed when bioluminescence intensity exceeded $5 \times 10^9$ photons per s.

## Lentiviral transduction of Raji_RFP_LUC cells
For lentiviral transduction of LUC–RFP vectors, 293T cells were seeded at 30% confluence in 10 cm dishes in DMEM 10% FBS and transfected the next day with the backbone of interest and the packaging plasmids pMD2.G and d8.9 using FuGENE HD (Promega). Media was changed 16 h after transfection. The viral supernatant was collected 24 and 48 h post transfection and incubated at 4 °C overnight with PEG800. It was then centrifuged at 3,500 r.p.m. for 1 h at 4 °C. The pellet was used to infect 200,000 Raji cells in the presence of 8 μg μl$^{-1}$ polybrene. Transduced Raji–RFP–LUC cells were sorted and maintained in RPMI 1640 with 10% FBS and 1% penicillin/streptomycin. Cells were then characterized in vitro for LUC expression levels and CD19 expression by flow cytometry.

## Immunofluorescence and immunohistochemistry staining
Immunohistochemical detection of CD19 was performed manually. After dewaxing and rehydration, sections were incubated for 10 min in 3% H$_2$O$_2$ in PBS to inhibit endogenous peroxidase. They were pretreated with 10 mM Tri Na citrate pH 6 for 20 min at 95 °C using PT module (Thermo Fisher Scientific). Slides were then blocked in 1% BSA in PBS for 30 min. Rabbit anti-human CD19 (rat anti-CD19, clone 6OMP31, eBioscience, catalogue number 14-0194-82) diluted 1:500 in 1% BSA was incubated overnight at 4 °C with agitation. After three washes in cold PBS, the secondary antibody rabbit (Thermo Fisher Scientific, catalogue number A-1107) diluted 1:1,000 in 1% BSA was applied for 30 min at room temperature. Sections were counterstained with DAPI and permanently mounted.

Analysis of CD3 (rabbit anti-CD3e, clone Sp7, Thermo Fisher, catalogue number MA5-14524, diluted 1:100) was performed using the fully automated Ventana Discovery ULTRA (Roche Diagnostics). All steps were performed on the machine with Ventana solutions. Briefly, dewaxed and rehydrated paraffin sections were pretreated with heat using standard condition (40 min) CC1 solution. The samples were incubated with the primary antibody for 1 h at 37 °C. After incubation with rabbit Immpress HRP (ready to use, Vector Laboratories), chromogenic revelation was performed with ChromoMap DAB kit (Roche Diagnostics). Sections were counterstained with Harris haematoxylin and permanently mounted. Slides were acquired with Leica DM5500 Upright Microscope and analysed using QuPath (protocol designed and performed by the EPFL Histology Core Facility).

## Statistical analysis
Statistical significance involving two groups was determined by two-tailed, unpaired Student's *t*-test. For comparison among three groups or more, ANOVA with multiple comparisons was used, and the *P* value was adjusted with Tukey's or Sidak's correction. Statistical significance in the Kaplan–Meier curve was determined using the Mantel–Cox log rank test. All *P* values were calculated using the GraphPad Prism software (v.9.1.2). In all graphs, error bars represent s.d.

## Reporting summary
Further information on research design is available in the Nature Portfolio Reporting Summary linked to this article.

# Data availability
All data generated in this study, including source data for the figures, are available from the corresponding author on reasonable request. Source data are provided with this paper.

# Code availability
Example data and scripts used for data curation, analysis and visualization are available at https://github.com/LSSI-ETH/Kapetanovic_2024.

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

## Acknowledgements

The authors thank the ETH Zurich Department of Biosystems Science and Engineering (D-BSSE) Single Cell Unit for their support and assistance throughout this study; École Polytechnique Fédérale de Lausanne (EPFL) Histology Core Facility for immunohistochemistry and chromogenic staining of tissue slides; Blutspendezenturm SRK Basel, University Hospital Basel, for the donor blood samples; A. Zippelius lab for providing the blinatumomab; and E. Aznauryan for his support early into this project. This work was supported by Helmut Horten Stiftung, Switzerland (to S.T.R.), National Center of Competence in Research (NCCR) Molecular Systems Engineering (to S.T.R. and R.J.P.) and Personalized Health and Related Technologies (to R.V.-L. and S.T.R).

## Author contributions

E.K. and S.T.R. designed the study; E.K., M.B., D.P., J.K., E.H., C.D., B.W., O.B., R.V.-L., K.K., L.R., K.L., R.K. and S.R. performed experiments. E.K., C.R.W. and M.B. analysed the data. R.B.D.R. and R.C.-R. assisted with obtaining biological material. P.S.D., R.J.P., E.O. and S.T.R. supervised the work. E.K. and S.T.R. wrote the manuscript with input from all authors.

## Funding

## Competing interests

E.K., S.T.R., C.R.W. and R.V.-L. are co-inventors on the patent 'Universal TCR variants for allogeneic immunotherapy' filed by ETH Zurich related to AED TCRs and methods of their use. The other authors declare no competing interests.

## Additional information

**Correspondence and requests for materials** should be addressed to Sai T. Reddy.

# Reporting Summary

## Statistics

For all statistical analyses, confirm that the following items are present in the figure legend, table legend, main text, or Methods section.

| n/a | Confirmed | |
|---|---|---|
| ☐ | ☒ | The exact sample size (*n*) for each experimental group/condition, given as a discrete number and unit of measurement |
| ☐ | ☒ | A statement on whether measurements were taken from distinct samples or whether the same sample was measured repeatedly |
| ☐ | ☒ | The statistical test(s) used AND whether they are one- or two-sided<br>*Only common tests should be described solely by name; describe more complex techniques in the Methods section.* |
| ☒ | ☐ | A description of all covariates tested |
| ☐ | ☒ | A description of any assumptions or corrections, such as tests of normality and adjustment for multiple comparisons |
| ☐ | ☒ | A full description of the statistical parameters including central tendency (e.g. means) or other basic estimates (e.g. regression coefficient) AND variation (e.g. standard deviation) or associated estimates of uncertainty (e.g. confidence intervals) |
| ☒ | ☐ | For null hypothesis testing, the test statistic (e.g. *F*, *t*, *r*) with confidence intervals, effect sizes, degrees of freedom and *P* value noted<br>*Give P values as exact values whenever suitable.* |
| ☒ | ☐ | For Bayesian analysis, information on the choice of priors and Markov chain Monte Carlo settings |
| ☒ | ☐ | For hierarchical and complex designs, identification of the appropriate level for tests and full reporting of outcomes |
| ☒ | ☐ | Estimates of effect sizes (e.g. Cohen's *d*, Pearson's *r*), indicating how they were calculated |

*Our web collection on statistics for biologists contains articles on many of the points above.*

## Software and code

Policy information about availability of computer code

| Data collection | IVIS Lumina II imaging system for in vivo mouse imaging. LSRFortessa (BD Biosciences) or a CytoFLEX(Beckman- Coulter) were used to collect flow-cytometry data. Cells were sorted using  BD FACSAria III or BD FACSAria Fusion instruments. Leica DM5500 Upright Microscope and Nikon Ti2 were used for microscopy imaging. |
|---|---|
| Data analysis | FlowJo 10, Geneious 11.1.4, Microsoft Excel for Mac, GraphPad Prism 9, QuPath 02.03, R (version 4.0.1.), MATLAB. |

For manuscripts utilizing custom algorithms or software that are central to the research but not yet described in published literature, software must be made available to editors and reviewers. We strongly encourage code deposition in a community repository (e.g. GitHub). See the Nature Portfolio guidelines for submitting code & software for further information.

## Data

Policy information about availability of data

All manuscripts must include a data availability statement. This statement should provide the following information, where applicable:

- Accession codes, unique identifiers, or web links for publicly available datasets
- A description of any restrictions on data availability
- For clinical datasets or third party data, please ensure that the statement adheres to our policy

All data generated in this study, including source data for the figures, are available as Source data or from the corresponding author on reasonable request.

## Research involving human participants, their data, or biological material

Policy information about studies with human participants or human data. See also policy information about sex, gender (identity/presentation), and sexual orientation and race, ethnicity and racism.

| | |
|---|---|
| Reporting on sex and gender | The study did not involve human participants. |
| Reporting on race, ethnicity, or other socially relevant groupings | – |
| Population characteristics | – |
| Recruitment | – |
| Ethics oversight | – |

Note that full information on the approval of the study protocol must also be provided in the manuscript.

# Field-specific reporting

Please select the one below that is the best fit for your research. If you are not sure, read the appropriate sections before making your selection.

☒ Life sciences ☐ Behavioural & social sciences ☐ Ecological, evolutionary & environmental sciences

For a reference copy of the document with all sections, see nature.com/documents/nr-reporting-summary-flat.pdf

# Life sciences study design

All studies must disclose on these points even when the disclosure is negative.

| | |
|---|---|
| Sample size | Statistical tests to predetermine sample size were not performed. We used 5 mice per experimental group (in the Raji-alone group we used 3 mice) to observe statistically meaningful differences between groups. |
| Data exclusions | No data were excluded from analysis. |
| Replication | The experimental results were replicated with several healthy human donors, and all attempts at replication were successful. |
| Randomization | For the in vivo experiments, mice were randomly distributed and injected with tumor cells and T cells. All experimental groups contained equal proportions of male and female animals. All mice were imaged on the same day of AED T/tumor cell injection, and tumor burden was consistent and reproducible across all mice. |
| Blinding | The researchers were not blinded, to ensure appropriate data acquisition and handling. |

# Reporting for specific materials, systems and methods

We require information from authors about some types of materials, experimental systems and methods used in many studies. Here, indicate whether each material, system or method listed is relevant to your study. If you are not sure if a list item applies to your research, read the appropriate section before selecting a response.

## Materials & experimental systems

| n/a | Involved in the study |
|---|---|
| ☐ | ☒ Antibodies |
| ☐ | ☒ Eukaryotic cell lines |
| ☒ | ☐ Palaeontology and archaeology |
| ☐ | ☒ Animals and other organisms |
| ☒ | ☐ Clinical data |
| ☒ | ☐ Dual use research of concern |
| ☒ | ☐ Plants |

## Methods

| n/a | Involved in the study |
|---|---|
| ☒ | ☐ ChIP-seq |
| ☐ | ☒ Flow cytometry |
| ☒ | ☐ MRI-based neuroimaging |

## Antibodies

| | |
|---|---|
| Antibodies used | The following antibodies were used in this study. From Biolegend: APC-CD3e(clone UCHT1 #300458) PE-Cy7- CD3e ( clone UCHT1, #300420), APC- CD4 (clone RPA-T4, #300552), PE-CD8a (clone HIT8a, #300908), PE-Cy7-CD19 (clone HIB19, #302216), PE-conjugated |

anti-human TCR α/β (clone IP26, #306707);PE anti-human CD107a (LAMP-1)antibody (clone H4A3, #328608); DAPI viability dye (Thermo Fisher, #62248);. Sytox Blue viability dye (Invitrogen, # S34857);CountBright Plus counting beads (Invitrogen, # C36995) .The following peptide-MHC dextramers were commercially obtained from Immudex: NY-ESO-1157-165 (SLLMWITQC, HLA-A*0201, #WB2696-PE); MART-126-35(27L) (ELAGIGILTV, HLA-A*0201, #WB2162-PE); MAGE-A3168-176 (EVDPIGHLY, HLA-A*0101, #WA3249-PE).

| Validation | All antibodies were validated by the respective manufacturer. |

## Eukaryotic cell lines

Policy information about cell lines and Sex and Gender in Research

| Cell line source(s) | Cell lines were originally obtained from ATCC. |

| Authentication | The cell lines were not authenticated beyond the ATCC provided certificates of analysis. Cell morphology and surface expression of antigens were consistent with published data. |

| Mycoplasma contamination | All cell lines were routinely tested for mycoplasma, and the results were negative. |

| Commonly misidentified lines (See ICLAC register) | No commonly misidentified cell lines were used. |

## Animals and other research organisms

Policy information about studies involving animals; ARRIVE guidelines recommended for reporting animal research, and Sex and Gender in Research

| Laboratory animals | NSG (NOD.Cg-PrkdcSCIDII2rgtm1Wjl/Sz) breeder pair mice were purchased from Charles River. Mice were bred in the EPFL mouse facility in accordance with IACUC regulations. Both female and male litter mates (5 weeks of age) were used for the experiments. |

| Wild animals | The study did not involve wild animals. |

| Reporting on sex | Both female and male litter mates (aged 5 weeks) were used in the experiments. |

| Field-collected samples | The study did not involve samples collected from the field. |

| Ethics oversight | All experiments were performed according to protocols approved by the EPFL IACUC. |

Note that full information on the approval of the study protocol must also be provided in the manuscript.

## Flow Cytometry

### Plots

Confirm that:

☒ The axis labels state the marker and fluorochrome used (e.g. CD4-FITC).

☒ The axis scales are clearly visible. Include numbers along axes only for bottom left plot of group (a 'group' is an analysis of identical markers).

☒ All plots are contour plots with outliers or pseudocolor plots.

☒ A numerical value for number of cells or percentage (with statistics) is provided.

### Methodology

| Sample preparation | For flow cytometry and cell sorting, cells were washed in (PBS 2% FBS) and incubated 20 minutes at 4 deg in FACS buffer (PBS 2% FBS, 2 mM EDTA) containing the desired antibodies. |

| Instrument | Cell sorting (FACS) was performed using BD FACSAria III or BD FACSAria Fusion instruments. |

| Software | FlowJo 10 software (BD Biosciences). |

| Cell population abundance | Flow cytometry was employed to ensure the purity of a target cell populations. |

| Gating strategy | FSC/SSC gating was used to define an initial lymphocyte gate together with the singlet gate. |

☒ Tick this box to confirm that a figure exemplifying the gating strategy is provided in the Supplementary Information.

