## [Peer Review File · Nature Biomedical Engineering]

Allogeneic T cells decoupling antigen binding and CD3 signalling enhance the antitumour activity of bispecific antibodies

Corresponding author: Sai Reddy

Editorial note

This document includes relevant written communications between the manuscript's corresponding author and the editor and reviewers of the manuscript during peer review. It includes decision letters relaying any editorial points and peer-review reports, and the authors' replies to these (under 'Rebuttal' headings). The editorial decisions are signed by the manuscript's handling editor, yet the editorial team and ultimately the journal's Chief Editor share responsibility for all decisions.

Any relevant documents attached to the decision letters are referred to as **Appendix #**, and can be found appended to this document. Any information deemed confidential has been redacted or removed. Earlier versions of the manuscript are not published, yet the originally submitted version may be available as a preprint. Because of editorial edits and changes during peer review, the published title of the paper and the title mentioned in below correspondence may differ.

Correspondence

Fri 21 Apr 2023

Decision on Article nBME-23-0599

Dear Dr Reddy,

Thank you again for submitting to *Nature Biomedical Engineering* your manuscript, "Enhancing bispecific antibody therapies by engineering allogeneic T cells via molecular decoupling of the TCR-CD3 signaling complex". The manuscript has been seen by three experts, whose reports you will find at the end of this message.

You will see that the reviewers appreciate the work, in particular from an immunology perspective. However, they express concerns about the degree of support for the claims, and provide useful suggestions for improvement. We hope that with further work you can address the criticisms and convince the reviewers of the merits of the study. In view of the comments of Reviewer #3 on the clinical implications of the work, please do amplify the discussion of the clinical relevance of the molecular decoupling of TCR signalling and CD3 signalling, adding any necessary caveats.

When you are ready to resubmit your manuscript, please upload the revised files, a point-by-point rebuttal to the comments from all reviewers, the reporting summary, and a cover letter that explains the main improvements included in the revision and responds to any points highlighted in this decision.

Please follow the following recommendations:

* Clearly highlight any amendments to the text and figures to help the reviewers and editors find and understand the changes (yet keep in mind that excessive marking can hinder readability).

* If you and your co-authors disagree with a criticism, provide the arguments to the reviewer (optionally, indicate the relevant points in the cover letter).* If a criticism or suggestion is not addressed, please indicate so in the rebuttal to the reviewer comments and explain the reason(s).

* Consider including responses to any criticisms raised by more than one reviewer at the beginning of the rebuttal, in a section addressed to all reviewers.

* The rebuttal should include the reviewer comments in point-by-point format (please note that we provide all reviewers will the reports as they appear at the end of this message).

* Provide the rebuttal to the reviewer comments and the cover letter as separate files.

We hope that you will be able to resubmit the manuscript within 16 weeks from the receipt of this message. If this is the case, you will be protected against potential scooping. Otherwise, we will be happy to consider a revised manuscript as long as the significance of the work is not compromised by work published elsewhere or accepted for publication at *Nature Biomedical Engineering*.

We hope that you will find the referee reports helpful when revising the work. Please do not hesitate to contact me should you have any questions.

Best wishes,

Pep

Pep Pàmies
Chief Editor, Nature Biomedical Engineering

Reviewer #1 (Report for the authors (Required)):

The authors in this research article engineer 'Allogenic-engineered-decoupled (AED)' T cells that rely on molecular decoupling of the TCR-CD3 complex. These T cells can bind to their specific antigen through their TCR, however, their signaling capacity is nullified due to a mutation in the TRAJ region. These cells still maintain CD3-mediated activation and can be used in conjugation with BiTE (blinatumomab) to clear tumor cells and minimize alloreactivity leading to a new combination therapy against cancer.

As an initial result, the authors showed that mutations previously identified in the TCRalpha connecting peptide that were supposed to reduce T cell responsiveness and not affect CD3-mediated activation lose their ability to respond to antigen and BiTE activation because of structural alterations in the complex due to the mutations. They test this by using DMF5 TCR, specific for MART-1 melanoma antigen, and a Jurkat-NFAT-GFP reporter system. Next, using sequence and structural analysis, they identified a novel conserved TCR motif – FGxGT present in TRAJ region that decouple TCR-antigen binding and CD3-mediated signaling. Then, the authors tested this mutation – FEQWT on 2 other TCRs – 1G4 and a3a which failed to produce the desirable effect due to poor surface expression. They subsequently created FGxGT libraries to derive variants that decouple antigen-based activation and BiTE-based CD3 activation. Further, the FEQWT mutant was tested in human primary cells that indicated that TCR-antigen interaction does not lead to T cell proliferation or cytokine secretion. Finally, the authors tested the activity of AED T cells in vivo in human tumor xenograft mouse model (Raji-RFP-Luc in NSG mice) and found that AED T cells in combination with blinatumomab cleared tumor cells.

Overall, the work indicates that molecular decoupling of antigen-based T cell activation in combination with CD3-targeting BiTE can lead to effective tumor response and no alloreactivity. In vitro and in vivo studies detailed in the results prove that this approach can be a viable cancer treatment option. Finally, the scientific work is thorough and moves the complicated TCR engineering field forward. However, no effort is made to understand the mechanistic details of the FEQWT mutation in the TCR-CD3 complex which is very important for publication in journals such as *Nature Biomedical Engineering*. The work as is feels more suited to core

immunology journals. The following major and minor concerns must be addressed.

Major concerns:

- 1) In line 119 and corresponding figure 1f, the authors state that they observed a significant drop in the TCR-CD3 surface expression and MART-1 dextramer binding with the 2 mutants tested. They could have sorted the wild type and mutants for comparable TCR and CD3 expressions and activated the complex with MART-1 dextramer and blinatumomab. This way they could conclusively establish whether antigen-mediated activation and CD3-mediated activation are decoupled or not. It should be noted that Brazin et al., reported near-equal expression of TCR and CD3 between the wild type and GGGSGSG mutant in their experiment.
- 2) In line 149, the authors indicate that they used structure-guided approach to design a candidate AED T cell variant: FEQWT mutant. However, no details are provided. Why is this mutant the best possible one?
- 3) From lines 253-263, the authors state that they use T2 cells as APCs for their co-culture experiments and monitor T2: T cell proportion using CD19 staining (Suppl figure 8d-f). Do T2 cells express CD19? Or the authors meant Raji cells here?
- 4) In figure 5c, only 1 mouse is shown in groups WT and WT+blina for day 35 and 42. This needs to be clarified.
- 5) Some mechanistic studies like in silico modeling or short molecular dynamics simulation to understand the effect of the mutation on the TCR will add value to this study especially since the same mutation on the TCRalpha subunit does not produce similar functional effect. Moreover, the same mutation on the TCRalpha subunit of 2 other TCRs – 1G4 and a3a lead to poor TCR-CD3 surface expression and blinatumomab activation. Data from mechanistic studies could explain this.

Minor changes:

- 1) Some labels in figures are tiny – Figure 1b, 3b.
- 2) The structure illustration in Figure 1g looks stretched.
- 3) In line 223 and line 224, the figure referenced should be Fig. 3c and Fig. 3d, respectively.

Reviewer #2 (Report for the authors (Required)):

The main goal of the study by Kapetanovic and colleagues is to engineer a solution for allogeneic T-cells which enables responsiveness to bispecific T-cell engagers (via the CD3e subunit) while precluding TCR signaling that may be alloreactive. The authors identified a conserved region (FGxGT) in the TRAJ region of TCR mutation of which leads to a marked decrease in T-cell activation upon exposure to a cognate pMHC but did not attenuate TCR crosslinking and degranulation by blinatumomab. The activity of the resulting AED T-cells was demonstrated using a MART-specific TCR in vitro and in vivo supporting feasibility of this approach for generation of non-alloreactive, off-the-shelf T-cell therapies that could serve as effectors with BiTEs.

The study sheds light into the molecular mechanics of TCR signaling illuminating the importance of the newly identified structural J-region domain in transmitting TCR signaling. However, several questions arise pertaining to the application of this technology in allogeneic T-cell therapies.

1. The authors argue AED T cells must retain the ability to bind cognate pMHC but do not explain why this is critical.
 - a. If FGxGT-mutated TCRs do not transmit the signal even upon binding its cognate ligand, why the specificity matters?
 - b. Could the authors simply delete TCR variable regions (TRAV/TRBV) or replace them with selection epitopes without mutating the FGxGT domain to achieve the same effect?
 - c. Would it be easier to use a well-characterized TCR known to be non-alloreactive, instead of mutating the TRAJ domain?

2. Attenuated responses of AED T cells to pMHC stimulation are very well demonstrated by the authors in various in vitro assays measuring proliferation and cytokine secretion. However, the authors do not show whether pMHC engagement is still enough to produce direct cell lysis. Have the cells completely lost the ability to kill target cells or there is residual cytotoxicity, even in the absence of proliferation or cytokine production?

3. The authors correctly identify a problem with BiTE MOA as they wear off endogenous T cells by promoting constant Signal 1 stimulation resulting in attrition and exhaustion. Would it be desirable if BiTEs exert the same effect on infused allogeneic T cells or it is better to protect those “elite soldiers” from BiTEs and let their activity and persistence be driven by CARs instead? Especially considering the infused allo-T would constitute only a minor fraction of the total T-cell content in the body.

Reviewer #3 (Report for the authors (Required)):

* A brief summary of the results.

The authors propose to generate an allogeneic T cell product to be used in combination with bispecific T cell engagers. By studying the TCR sequence and structure the identity a couple of regions of interest, and show that altering the TRAJ FGxGT motif can lead to the good surface expression of multiple TCRs, that are still capable of binding their cognate antigens but fails to activate T cells. The activation by bispecific T cell engagers is unaltered. They also show that these T cells are able to control the tumor in presence of blinatumomab in an in vivo mouse model.

* Your reasoned opinion on the degree of advance (fundamental, mechanistic, methodological, technological, performance-wise, translational and/or clinical) of the work with respect to the state of the art. If the results or conclusions are not original, please provide relevant references.

The research presented here is novel, as it touches on the requirements of TCR signal transduction. This makes it an interesting paper for both fundamental TCR immunologist as translational T cell product developers. However, I am really wondering what the broader implication of this strategy is. In our daily clinical practice T cell penia is not a substantial problem also post allo SCT (see eg Gaballe et al. Blood 2022) and main impact for efficacy of bispecific seg post allo SCT was the presence of memory immune cells being present. Also no substantial toxicity was observed within the context, implying that the indicated clinical challenge as outlined in the discussion seems not to be a hurdle. In a nutshell, authors present a solution for a clinical problem that does not exist, Thus overall, although interesting mutations I consider the clinical impact as rather low, as complete engineering of immune cells like CAR T from third part with additional engineering might be more suitable.

* Your reasoned opinion on the implications of the findings.

The strategy designed by the authors is very interesting, the biological point of view, but in my perception not solving a major clinical problem.

As numbered lists:

* Any major technical criticisms or questions.

1. Throughout the paper there is a huge discrepancy in experimental setup between the assessment of DMF5 TCR mediated activation and CD3 (blinatumomab) mediated activation. For the TCR mediated activation the authors titrate the MART-1 peptides over 4 orders of magnitude, resulting in a clear dose response curve. This has not been done with blinatumomab, the 2-fold dilution series covers 3 ng/ml to 12 ng/ml, resulting in full activation for all conditions with the WT TCR. It would be more informative than the authors used a wider range of blinatumomab concentration that also covers a concentration that induces half maximum T cell activation. Mutations introduced in the FGxGT, might induce more subtle changes in responsiveness that are not captured using the current blinatumomab concentrations. (examples fig 2f-g and fig 3c-e)

2. The structure guided approach for introducing glutamic acid and tryptophan in the FGxGT motif is ill-defined. Why using these two amino acids and not other charged or bulky residues? The structure based rationale is missing.

See lines 149-152: "We used a structure-guided approach to design a candidate AED T cell variant: the mutated motif (FEQWT) was incorporated into the backbone of the DMF5 TCR and integrated via Cas9-mediated HDR into the genome of Jurkat cells (Jkt-AEDDMF5 01) (Supplementary Fig. 1a)."

3. For showing the potential of the AED T cells towards clinical application the authors use an in vivo experiment that is far from clinical reality, where tumors are surrounded by 15x more T cells. We acknowledge the fact that setting up a more clinical relevant model for bispecific T cell engagers is challenging, but it would be interesting to see if similar results are obtained when T cells are not mixed with tumor cells, to mimic the patient setting (e.g. as done here 10.1038/s41408-022-00634-4). Also the control group used in the in vivo experiment, a mix of T cells of 2 donors is something that will not be done in the clinic.

4. In the same line of 3, the authors state that elimination of the tumor in the absence of engineering results in tumor control due to allo reactivity. However the authors do not report on mice dying from GVHD, please clarify.

* Any minor technical criticisms or questions.

1. The conserved FGxG motif in the TRAJ gene segment is presented as a novel finding, but it has been one of the key elements to identify J-segments of both TCR and IG. Moreover there are several studies that imply that this motif is structurally affected by peptideHLA recognition of the TCR (10.1016/j.immuni.2011.04.017 and 10.1074/jbc.RA118.003832). The authors should acknowledge these prior findings within the manuscript.

2. In Lines 211-220 the authors discuss the selection of clones with an altered FGxGT motif based on the difference in Rank between the Peptide activated T cells and Blinatumomab activated T cells. However, it is not clear whether one can make a clear selection based on the enrichment of certain mutations in the different FACS populations.

E.g. the FEQWT motif is highly enriched in all populations for both DMF5 and 1G4, and decently enriched for all populations for TCR A3A (Figure S6a-c). Would the authors have picked this FEQWT motif, used in AEDDMF5 01, based on their screening method?

Moreover the library selection rounds, depicted in figure S5d seems to select for an opposite effect, from left to right the percentage of the T cells that are active against the peptide increase, while the blinatumomab reactive T cells decrease in percentage. Could the authors comment on this?

3. Did the authors observe any differences between CD8 and CD4 cells in the experiments with primary T cells?

* Any missing or unclear details about statistics, protocols or materials (please check the reporting summary provided; this is a dynamic PDF file that when not flattened can only be properly viewed via Acrobat Reader).

* Any missing citations to relevant literature.

* Any optional suggestions for improvement.

* Any stylistic issues or recommendations.

N/A

Tue 13 Jun 2023

Decision on Article nBME-23-0599A

Dear Dr Reddy,

Thank you for your revised manuscript, "Enhancing bispecific antibody therapies by engineering allogeneic T cells via molecular decoupling of the TCR-CD3 signaling complex", which has been seen by the original reviewers. In their reports, which you will find at the end of this message, you will see that the reviewers acknowledge the improvements to the work, that Reviewer #2 asks for evidence on the level of target-cell killing, and that Reviewer #1 presses on a few points, in particular about the importance of providing some mechanistic evidence, that we hope you will be able to address in a further revised manuscript.

As before, when you are ready to resubmit your manuscript, please upload the revised files, a point-by-point rebuttal to the comments from all reviewers, the reporting summary, and a cover letter that explains the main improvements included in the revision and responds to any points highlighted in this decision.

As a reminder, please follow the following recommendations:

- * Clearly highlight any amendments to the text and figures to help the reviewers and editors find and understand the changes (yet keep in mind that excessive marking can hinder readability).
- * If you and your co-authors disagree with a criticism, provide the arguments to the reviewer (optionally, indicate the relevant points in the cover letter).
- * If a criticism or suggestion is not addressed, please indicate so in the rebuttal to the reviewer comments and explain the reason(s).
- * Consider including responses to any criticisms raised by more than one reviewer at the beginning of the rebuttal, in a section addressed to all reviewers.
- * The rebuttal should include the reviewer comments in point-by-point format (please note that we provide all reviewers with the reports as they appear at the end of this message).
- * Provide the rebuttal to the reviewer comments and the cover letter as separate files.

We hope that you will be able to resubmit the manuscript within 12 weeks from the receipt of this message. If this is the case, you will be protected against potential scooping. Otherwise, we will be happy to consider a revised manuscript as long as the significance of the work is not compromised by work published elsewhere or accepted for publication at *Nature Biomedical Engineering*.

We look forward to receive a further revised version of the work. Please do not hesitate to contact me should you have any questions.

Best wishes,

Pep

Pep Pàmies
Chief Editor, Nature Biomedical Engineering

Reviewer #1 (Report for the authors (Required)):

Based on the revised manuscript, there are still critical questions that remain unanswered.

1. The authors claim their approach as 'structure-based' to identify a mutant that could decouple signal

transmission from antigen engagement. However, they did not answer the question that why FEQWT is the best mutant for this approach? At the least they should screen for more mutants to identify the best possible mutant.

2. It is hard to envision the universality of this approach as the same mutation on 2 other TCRs resulted in poor expression and blinatumomab activation.

3. Further, no mechanistic details about the workings of this mutation through MD and modeling are provided. The work is more suited to core immunology journals.

Reviewer #2 (Report for the authors (Required)):

The authors addressed most of my queries. I have two points remaining:

1. Answer to 1c should be incorporated into the manuscript like the preceding points.

2. Measuring the level of target cell killing would be very important for this study. Different effector functions (proliferation, cytokine secretion, cytolysis) require different levels of TCR stimulation, and avoiding unwanted tissue toxicity is very important for this conceptual study. These assays are easy to do.

Reviewer #3 (Report for the authors (Required)):

no further comments

Mon 13 Nov 2023

Decision on Article nBME-23-0599B

Dear Dr Reddy,

Thank you for your revised manuscript, "Enhancing bispecific antibody therapies by engineering allogeneic T cells via molecular decoupling of the TCR-CD3 signaling complex". Having consulted with Reviewers #1 and #2 (who have no further comments), I am pleased to write that we shall be happy to publish the manuscript in *Nature Biomedical Engineering*.

We will be performing detailed checks on your manuscript, and in due course will send you a checklist detailing our editorial and formatting requirements. You will need to follow these instructions before you upload the final manuscript files.

Best wishes,

Pep

Pep Pàmies
Chief Editor, Nature Biomedical Engineering

Reviewer #1 (Report for the authors (Required)):

The authors have addressed the previous issues. No further comments

Reviewer #2 (Report for the authors (Required)):

I have no remaining comments

Rebuttal 1

Reviewer #1

R1.1. In line 119 and corresponding figure 1f, the authors state that they observed a significant drop in the TCR-CD3 surface expression and MART-1 dextramer binding with the 2 mutants tested. They could have sorted the wild type and mutants for comparable TCR and CD3 expressions and activated the complex with MART-1 dextramer and blinatumomab. This way they could conclusively establish whether antigen-mediated activation and CD3-mediated activation are decoupled or not. It should be noted that Brazin et al., reported near-equal expression of TCR and CD3 between the wild type and GGGSGSG mutant in their experiment.

→We appreciate the reviewer's attention to the TCR-CD3 expression characteristics. In the study, all of our engineered T cell clones (DMF5_{FATADALN}, DMF5_{GGGSGSG}, DMF5 WT, and AED_{DMF5}) were sorted and a monoclonal population was isolated prior to functional assays being performed (antigen stimulation or blinatumomab activation). However, due to the introduced mutations, DMF5_{FATADALN} and DMF5_{GGGSGSG} variants did not possess equal levels of TCR expression relative to WT, which was evident in their CD3 expression and binding to the MART-1 dextramer (Fig. 1f).

R1 also comments on the Brazin et al. study, which was performed exclusively with mouse T cells and TCRs. A previous study by Steven Rosenberg and colleagues (Cohen et al., Cancer Res 2006; doi: [10.1158/0008-5472.CAN-06-1450](https://doi.org/10.1158/0008-5472.CAN-06-1450)) reported that mouse TCRs have higher TCR-chain pairing ability relative to human T cells/TCRs and thus express better even when mutations in the chains are introduced. The following quote from Cohen et al. provides a clear explanation:

*“Importantly, we show that TCR with mouse constant regions functions better in human cells than its human counterpart, leading to an increased sensitivity to tumor cells. Biochemical analysis suggested that part of this enhanced activity was due to the **preferential pairing of murine constant regions** with themselves and less mispairing with the endogenous human TCR chains and to **increased stability** of the TCR/CD3 ζ complex.”*

We have addressed this point of mouse TCR stability in the revised manuscript text and have cited Cohen et al. (citation 26, pg. 4, lines 123-125).

R1.2. In line 149, the authors indicate that they used a structure-guided approach to design a candidate AED T cell variant: FEQWT mutant. However, no details are provided. Why is this mutant the best possible one?

Related comment from R3:

R3.2. The structure guided approach for introducing glutamic acid and tryptophan in the FGxGT motif is ill-defined. Why using these two amino acids and not other charged or bulky residues? The structure based rationale is missing.

See lines 149-152: “We used a structure-guided approach to design a candidate AED T cell variant: the mutated motif (FEQWT) was incorporated into the backbone of the DMF5 TCR and integrated via Cas9-mediated HDR into the genome of Jurkat cells (Jkt-AEDDMF5 01) (Supplementary Fig. 1a).”

→ We thank R1 and R3 for the request of a more detailed explanation on how the structure-oriented mutation has been derived. Further, we thank the R3 for pointing out the important work of Yin et al. (now cited in the manuscript), which described the FGxGT as a highly conserved motif and suggested that the motif acts as a swivel, a flex point adjusting the association of Va and Vb domains. Yin et al. identified the 2nd glycine (G102) of the FGxGT motif in the J elements as the responsible residue that may affect how CDRs align and interact with the individual MHC molecules.

In our study, we hypothesize that the FGxGT motif could also be responsible for transmitting the signal generated by peptide binding to CD3 molecules. Our engineering efforts aimed to introduce mutations into the Jkt-DMF5 receptor that would preserve TCR stability, disrupt the association of the alpha and beta chains, and cease the natural signal transmission upon peptide-MHC engagement. Since glycine residues are often found in flexible regions (and synthetic linkers) of proteins due to their small size and ability to adopt multiple conformations, to disrupt the suspected function of the motif and at the same time preserve receptor stability we replaced the first glycine residue (G100) in the alpha chain FGxGT with glutamate (E), a large and negatively charged amino acid that we hypothesized would disrupt the flexibility in the region and the Va-Vb interchain association. Further, the pivotal and second glycine (G102) of FGxGT was replaced with tryptophan (W) to stabilize the alpha chain, as aromatic residues are favored in beta-sheet stabilization.

We have now added an explanation for the rationale behind our structure-guided approach to engineering the FGxGT motif in the revised manuscript (see citation 18, pg. 6, lines 159-172).

R1.3. From lines 253-263, the authors state that they use T2 cells as APCs for their co-culture experiments and monitor T2: T cell proportion using CD19 staining (Suppl figure 8d-f). Do T2 cells express CD19? Or the authors meant Raji cells here?

→The T2 cell line is a negative mutant for “Transporter associated with antigen processing” (TAP protein), expressing an empty HLA-A*0201 allele of the MHC-I c on the cell surface. The T2 cell line has been derived from lymphoblasts and thus does indeed express CD19, which we utilized to distinguish T2 cells and human T cells in flow cytometry.

R1.4. In figure 5c, only 1 mouse is shown in groups WT and WT+blina for day 35 and 42. This needs to be clarified.

→In the manuscript text (pg 13, lines 316-318), we state that several mice were sacrificed due to the loss of tumor growth in the WT T cell _ Raji cells cohort. This is the reason why there was only one mouse shown for these groups in Figure 5c. For additional clarification, we have now added the number of mice that have been sacrificed (4) in the results section (pg. 14, line 343-345).

R1.5. Some mechanistic studies like in silico modeling or short molecular dynamics simulation to understand the effect of the mutation on the TCR will add value to this study especially since the same mutation on the TCRalpha subunit does not produce similar functional effect. Moreover, the

same mutation on the TCRalpha subunit of 2 other TCRs – 1G4 and a3a lead to poor TCR-CD3 surface expression and blinatumomab activation. Data from mechanistic studies could explain this.

→While we agree that the addition of molecular dynamics simulation and in silico modeling would indeed be interesting, these types of analysis are outside our area of expertise. Further, since the transduction of signaling in TCRs is not well understood, and there are no significant structural changes observed between MHC bound and unbound TCRs, we are uncertain if we would be able to discern the change in function through these models. For this proof of concept, translational immunology study, we firmly believe that functional screening and analysis are most important. We do agree with the reviewer though that future studies incorporating molecular dynamics and in silico modeling would add value for elucidating the mechanistic effect of mutations on AED T cells and have added a short statement in the Discussion to propose this (pg.16 lines 377-378).

R1.1 minor comments:

1) Some labels in figures are tiny – Figure 1b, 3b.

Figures adjusted (please see pg. 3 and 9)

2) The structure illustration in Figure 1g looks stretched.

Figure adjusted (please see pg. 9)

3) In line 223 and line 224, the figure referenced should be Fig. 3c and Fig. 3d, respectively.

Corrected (please see pg.10 now lines 251 and 252)

Reviewer #2

R2.1. The authors argue AED T cells must retain the ability to bind cognate pMHC but do not explain why this is critical.

→We thank R2 for his comment, and this has now been addressed in the manuscript (pg. 5- 6 and 17) as follows:

a. If FGxGT-mutated TCRs do not transmit the signal even upon binding its cognate ligand, why the specificity matters?

→The main reason we wish to have the AED TCR retain the ability to bind the cognate pMHC is that it maintains the same specificity/cross-reactivity profile. A TCR that has been mutated and loses specificity to its cognate pMHC would not necessarily mean that the TCR is non-reactive, but rather that we have lost the information on its specificity and it could now be specific/cross-reactive for an entirely different peptide-MHC. Engineering TCR specificity might introduce unforeseen cross-reactivity and have severe consequences, for example, in a previous T cell therapy clinical trial, a TCR engineered for enhanced affinity to a tumor-associated antigen

(MAGE-A3 peptide) showed an unexpected cross-reactivity toward a self-antigen expressed by beating cardiomyocytes, which ultimately resulted in treatment-induced patient deaths (Cameron et al., 2013; Linette et al., 2013). Therefore it is critical for any TCR-based cell therapy to have a very precisely defined specificity and cross-reactivity profile. Furthermore, we validate that TCR - CDR3 decoupling has been achieved by maintaining specificity for cognate p-MHC, rather than a change in the specificity/cross-reactivity profile.

We have now added additional text to the Results section to further clarify why maintaining the specificity of the TCR to cognate peptide-MHC is critical (see pg. 5, lines 143-148).

b. Could the authors simply delete TCR variable regions (TRAV/TRBV) or replace them with selection epitopes without mutating the FGxGT domain to achieve the same effect?

→The approach R2 suggests would be much simpler and is something we initially explored. However, our findings (as well as others in the field) have shown that deletion/knockout of TCR variable regions does not appear to be effective, as the deletion of any of the components of the TCR receptor would lead to a disruption in chain folding, association, and ultimately the surface expression of the CD3 co-receptor. The individual components of the TCR/CD3 complex would be marked for degradation and no functional CD3 receptor would be available on the T cell for bispecific antibodies to engage. Indeed, we demonstrate this in Figure 1a, as a disruption of the TCR variable alpha gene by CRISPR-Cas9 led to a complete loss of surface expression of CD3.

This has been added in the text (see pg. 4, lines 136-138)

c. Would it be easier to use a well-characterized TCR known to be non-alloreactive, instead of mutating the TRAJ domain?

→This strategy would be an option if the MHC configuration of the patient and donor are fully matched. However, proper MHC matching is a difficult task to achieve and even in studies with fully matched related donors, the occurrence of life-threatening GvHD in hematopoietic stem cell transplantation (HSCT) is as high as 50% (Srinagesh et al; Ther Adv Hematol. 2019; doi: 10.1177/2040620719891358). Further, no TCR is by default a non-alloreactive TCR as in the context of a given peptide and MHC allele, any TCR could become cross-reactive, this is why further receptor engineering, such as AED, is necessary to achieve safety in adoptive T cell transfers.

R2.2. Attenuated responses of AED T cells to pMHC stimulation are very well demonstrated by the authors in various in vitro assays measuring proliferation and cytokine secretion. However, the authors do not show whether pMHC engagement is still enough to produce direct cell lysis. Have the cells completely lost the ability to kill target cells or there is residual cytotoxicity, even in the absence of proliferation or cytokine production?

→In this study, we did not directly examine cell lysis, however, the principle of T cell activation and overall response is well established to be based on T cell proliferation, differentiation, and cytokine production. Particularly, T cell proliferation has been used as a key measure to determine alloreactivity (e.g. in mixed lymphocyte reaction (MLR)). Further, the release of cytokines is directly linked to the cytotoxic effect, and given the attenuated profile of AED T cells across an extensive peptide range, in terms of cytokines and overall proliferation, any residual cytotoxicity, independent of proliferation and cytokine release, is highly unlikely.

R2.3. The authors correctly identify a problem with BiTE MOA as they wear off endogenous T cells by promoting constant Signal 1 stimulation resulting in attrition and exhaustion. Would it be desirable if BiTEs exert the same effect on infused allogeneic T cells or it is better to protect those “elite soldiers” from BiTEs and let their activity and persistence be driven by CARs instead? Especially considering the infused allo-T would constitute only a minor fraction of the total T-cell content in the body.

→The field of bispecific antibodies is rapidly evolving, including the engineering of more complex multi-specifics/ combination bispecific antibodies that in addition to targeting CD3, also target other co-stimulatory receptors on T cells (e.g., 41BB/CD137, CD28, see here for example <https://doi.org/10.1126/scitranslmed.aaw7888>). This strategy addresses the need for co-stimulatory signals to achieve sustained and improved T-cell engagement. However, the question of CAR molecules runs into the problem of persistent antigen stimulation much like the endogenous T cells, which can drive T cells into exhaustion (Soerens et al.; Nature 2023; doi.org: 10.1038/s41586-022-05626-9).

A combination approach of healthy AED T cells with improved bispecific engagers could have a significant clinical impact as AED T cells could be, in principle, stimulated multiple times and even easily switch targets as the tumor antigen landscape changes. Further, T cells can rest in between stimulation, which would likely increase their longevity and the duration of the therapeutic effect.

To clarify these points and provide the direct context of T cell exhaustion and comparison to CARs, please see the revised manuscript Discussion section (pg. 17, lines 420-429).

Reviewer #3

R3.1 Throughout the paper there is a huge discrepancy in experimental setup between the assessment of DMF5 TCR mediated activation and CD3 (blinatumomab) mediated activation. For the TCR mediated activation the authors titrate the MART-1 peptides over 4 orders of magnitude, resulting in a clear dose response curve. This has not been done with blinatumomab, the 2-fold dilution series covers 3 ng/ml to 12 ng/ml, resulting in full activation for all conditions with the WT TCR. It would be more informative than the authors used a wider range of blinatumomab

concentration that also covers a concentration that induces half maximum T cell activation. Mutations introduced in the FGxGT, might induce more subtle changes in responsiveness that are not captured using the current blinatumomab concentrations. (examples fig 2f-g and fig 3c-e)

→ We thank the reviewer for this valuable suggestion, to address this we have performed additional experiments over an extensive range of blinatumomab concentrations. The additional experimental results showcase that AED_{DMF501} variant performs equally well when activated with both low and high blinatumomab concentrations compared to the WT DMF5, thus confirming AED T cells do not have a higher threshold of activation.

Please see the revised text (pg. 7, lines 190-192) and Supplementary Figure 3c).

R3.2. The structure guided approach for introducing glutamic acid and tryptophan in the FGxGT motif is ill-defined. Why using these two amino acids and not other charged or bulky residues? The structure based rationale is missing.

See lines 149-152: “We used a structure-guided approach to design a candidate AED T cell variant: the mutated motif (FEQWT) was incorporated into the backbone of the DMF5 TCR and integrated via Cas9-mediated HDR into the genome of Jurkat cells (Jkt-AEDDMF5 01) (Supplementary Fig. 1a).”

Related comment from R1: (as before)

R1.2. In line 149, the authors indicate that they used a structure-guided approach to design a candidate AED T cell variant: FEQWT mutant. However, no details are provided. Why is this mutant the best possible one?

→ We thank R1 and R3 for the request of a more detailed explanation on how the structure-oriented mutation has been derived. Further, we thank the R3 for pointing out the important work of Yin et al. (now cited in the manuscript), which described the FGxGT as a highly conserved motif and suggested that the motif acts as a swivel, a flex point adjusting the association of Va and Vb domains. Yin et al. identified the 2nd glycine (G102) of the FGxGT motif in the J elements as the responsible residue that may affect how CDRs align and interact with the individual MHC molecules.

In our study, we hypothesize that the FGxGT motif could also be responsible for transmitting the signal generated by peptide binding to CD3 molecules. Our engineering efforts aimed to introduce mutations into the Jkt-DMF5 receptor that would preserve TCR stability, disrupt the association of the alpha and beta chains, and cease the natural signal transmission upon peptide-MHC engagement. Since glycine residues are often found in flexible regions (and synthetic linkers) of proteins due to their small size and ability to adopt multiple conformations, to disrupt the suspected function of the motif and at the same time preserve receptor stability we replaced the first glycine residue (G100) in the alpha chain FGxGT with glutamate (E), a large and negatively charged amino acid that we hypothesized would disrupt the flexibility in the region and the Va-Vb interchain association. Further, the pivotal and second glycine (G102) of FGxGT was replaced with tryptophan (W) to stabilize the alpha chain, as aromatic residues are favored in beta-sheet stabilization.

We have now added an explanation for the rationale behind our structure-guided approach to engineering the FGxGT motif in the revised manuscript (see citation 18, pg. 6, lines 159-172).

R3.3 For showing the potential of the AED T cells towards clinical application the authors use an in vivo experiment that is far from clinical reality, where tumors are surrounded by 15x more T cells. We acknowledge the fact that setting up a more clinical relevant model for bispecific T cell engagers is challenging, but it would be interesting to see if similar results are obtained when T cells are not mixed with tumor cells, to mimic the patient setting (e.g. as done here 10.1038/s41408-022-00634-4). Also the control group used in the in vivo experiment, a mix of T cells of 2 donors is something that will not be done in the clinic.

→With our in vivo experiment we show a proof-of-concept that AED T cells can effectively clear tumor cells without unwanted allo-response. We have used a similar experiment as previously performed by Dreier et al. (Int J Cancer 2002; DOI: 10.1002/ijc.10557) who showed initial blinatumomab activity, however in their study they used a much higher E: T ratio of 100:1 (compared to our study of 15:1). We agree that in future pre-clinical studies, the systemic model would be interesting and valuable, however, such a mouse model is beyond the scope of this paper. As R3 has stated these in vivo models will be difficult to establish, and even in the study R3 references (Lei et al, Blood Cancer Journal, 2022), the experimental setup exceeds our 15:1 ratio, as Lei et al used 100 times more T cells than Raji cells (1×10^7 cells T cells vs. 1×10^5 Raji). In addition, Lei et al used in vivo antitumor efficacy experiments consisting of 10^7 T- cells introduced in mice every other day, which far exceeds the number of T cells we have used per mouse (and is feasible to generate in an academic study).

We do agree with the R3 that currently mixing of T cells of different donors is not a feasible clinical strategy, but with our engineering of AED T cells that are safe from alloreactivity and GvHD, we have a proof-of-concept to demonstrate that this may be feasible in the future.

R3.4. In the same line of 3, the authors state that elimination of the tumor in the absence of engineering results in tumor control due to allo reactivity. However the author do not report on mice dying from GVHD, please clarify.

→In the cohort consisting of WT non-engineered donor cells and Raji cells, Raji cells were cleared in the absence of targeted blinatumomab activation. This is due to an allo-response of donor T cells against Raji cells. The GvHD against mice has thus not occurred.

This has now been stated more clearly in the manuscript (pg. 14, lines 343-344; pg. 15, lines 354-355).

R3.5 The research presented here is novel, as it touches on the requirements of TCR signal transduction. This makes it an interesting paper for both fundamental TCR immunologist and translational T cell product developers. However, I am really wondering what the broader

implication of this strategy is. In our daily clinical practice T cell penia is not a substantial problem also post allo SCT (see eg Gaballe et al. Blood 2022) and the main impact for efficacy of bispecific seg post allo SCT was the presence of memory immune cells being present. Also no substantial toxicity was observed within the context, implying that the indicated clinical challenge as outlined in the discussion seems not to be a hurdle. In a nutshell, authors present a solution for a clinical problem that does not exist, Thus overall, although interesting mutations I consider the clinical impact as rather low, as complete engineering of immune cells like CAR T from third part with additional engineering might be more suitable.

→ We thank R3 for finding our work novel and important both in the context of fundamental TCR biology and the development of translational cell therapies.

The pre-conditioning of patients for the allogeneic hematopoietic stem cell transplantation (HSCT) is harmful to the immune system, and the first 100 days after HSCT (engraftment phase) is characterized by cellular immunodeficiencies, due to a reduced number of natural killer (NK) cells of the innate immune system and T cells of the adaptive immune system. (Ogonek et al.; Frontier in Immunology 2016.; doi: 10.3389/fimmu.2016.00507). Thus, the combination therapy of healthy donor T cells and bispecific antibodies could benefit patients with early post-HSCT relapse. In a similar way, many patients might not have cytopenia per se, but the functionality of their immune system due to previous rounds of chemotherapy is greatly impaired, which is why patients are prone to different kinds of infections, including the reactivation of cytomegalovirus (CMV) and Epstein–Barr virus (EBV).

Further, the clinical efficacy of immunotherapy (CAR- T, TCR-T, bispecific antibodies), depends on T cell number and functionality. The need for a healthy and sufficient number of T cells has been clearly demonstrated in a previous publication conducted by Genentech researchers on mosunetuzumab, a clinically approved anti-CD20 bispecific antibody:

Sun et al., **Anti-CD20/CD3 T cell–dependent bispecific antibody for the treatment of B cell malignancies** Sci Transl Med 2015 (doi: 10.1126/scitranslmed.aaa4802).

An important piece of data is presented in Fig. 6B and is described by the following statement:

“As expected, we found that T cell content varied substantially (between 0.4 and 8% of mononuclear cells). Strikingly, we observed that the extent of tumor cell killing was highly correlated to T cell content (R2 value of 0.9, Fig. 6A)...we tested whether the addition of healthy donor T cells could enhance CD20-TDB activity. This was indeed the case, demonstrating that T cell content of the cultures was the primary determinant of TDB (T cell-dependent bispecific antibody) activity in these cultures (Fig. 6B).”

In addition, the key finding of the study R3 refers to (Gaballa et al.; Blood 2022.; <https://doi.org/10.1182/blood.2021013290>) clearly states that the differentiating element between responders and non-responders to blinatumomab post-HSCT is the patient’s immune milieu at the time of the treatment.

“ The composition of a patient’s T-cell subsets at the time of treatment is indicative of whether they will respond to blinatumomab.”

Very recently in 2023, a high-profile and relevant important study was published:

Friedrich et al. **The pre-existing T cell landscape determines the response to bispecific T cell engagers in multiple myeloma patients** ; Cancer Cell 2023 (<https://doi.org/10.1016/j.ccell.2023.02.008>).

Friedrich et al. was published online a few weeks after our original manuscript submission and was thus not initially cited. In this study, the authors characterize another bispecific antibody, teclistamab (anti-BCMA x anti-CD3), where it was again demonstrated that the difference between patient non-responders and responders lies in the particular subsets and numbers of T cells.

Collectively, these translational studies support the fact that the clinical efficacy of bispecific antibody therapies depends on the quality and quantity of patient T cells. These results are shown in the context of both in vitro and in vivo data, including with clinically developed or approved molecules in different patient subtypes and across different cancer indications. Therefore, we feel this corroborates the central motivation of our manuscript and technology that there is clinical potential for augmenting bispecific antibody therapy through the addition of functional, healthy T cells.

We appreciate that these points may not have been as clearly explained in the initial manuscript, therefore we have revised the intro and discussion sections, including adding citations, please see **pg. 1, lines 42-46, and pg. 17. lines 426-429, citations 8, 42 and 48.**

We do not view AED T cells as direct competition for CAR-T cell therapy, as is the case for many cancer patients, there is a need for more options. Especially since different patient subsets with varying cancer indications respond favorably to different treatment conditions.

Finally, it is important to note that combination therapy of AED T cells + biAbs could have several important advantages. In principle, multiple cancer antigens can be targeted simultaneously or consecutively as the tumor antigen landscape changes, and therapy can be switched on and off, which would allow T cells to rest and have a prolonged therapeutic effect. The potential to combine AED T cells with a myriad of CD3 engagers currently in pre-clinical and clinical development could bring a substantial clinical benefit to patients. Indeed, we demonstrate in the revised manuscript that in addition to the CD19-targeting blinatumomab, AED T cells are compatible with other clinically approved bispecific antibodies against a different target (CD20) and in a different format, please see **new Supp Fig 3d.**

* Any minor technical criticisms or questions.

R3.1. minor The conserved FGxG motif in the TRAJ gene segment is presented as a novel finding, but it has been one of the key elements to identify J-segments of both TCR and IG. Moreover there are several studies that imply that this motif is structurally affected by peptideHLA recognition of the TCR ([10.1016/j.immuni.2011.04.017](https://doi.org/10.1016/j.immuni.2011.04.017) and

10.1074/jbc.RA118.003832). The authors should acknowledge these prior findings within the manuscript.

→We thank R3 for this remark, and this work has now been properly cited in the manuscript citations 17 and 18 pg. 2, lines 70-73, pg. 6, lines 159-163, and pg. 16 lines 373-377.

2. In Lines 211-220 the authors discuss the selection of clones with an altered FGxGT motif based on the difference in Rank between the Peptide activated T cells and Blinatumomab activated T cells. However, it is not clear whether one can make a clear selection based on the enrichment of certain mutations in the different FACS populations.

E.g. the FEQWT motif is highly enriched in all populations for both DMF5 and 1G4, and decently enriched for all populations for TCR A3A (Figure S6a-c). Would the authors have picked this FEQWT motif, used in AEDDMF5 01, based on their screening method?

→What we have followed in our variant selection is the dynamic ranking as the variants move through the functional screening since the absolute gain/loss in rank is very much driven by the initial variant distribution and ranking. Further, the intrinsic property of T cells, and thus our functional assays, is that only a fraction of T cells (~30%), even in a monoclonal Jurkat population, is activated at a time. Given the incomplete activation of T cells, and the difficulty to remove the WT variant from the peptide-negative pool, each library was started with the FEQWT variant that had previously been found to be decoupled in Jrkt-AED_{DMF5 01} but was however difficult to express in the a3a and 1G4 TCR backgrounds. This is why in the library NGS data this variant is overrepresented, and would thus not be selected from the library screening. However, this design allowed for a more robust exclusion of peptide-responsive variants.

This has now been better explained in the following part of the manuscript, please see pg. 8 lines 222-226.

Moreover the library selection rounds, depicted in figure S5d seems to select for an opposite effect, from left to right the percentage of the T cells that are active against the peptide increase, while the blinatumomab reactive T cells decrease in percentage. Could the authors comment on this?

→This is an effect of two issues:

1. The experimental noise coming from the biological samples as Jurkat T-cells do not always have the same activation signal, thus the percentage of cells can vary.
2. As from round to round we are “cleaning” the library from the peptide reactive clones, as such the percentage of blinatumomab activation decreases, as most of the variants in the library that are peptide responsive are very likely to also be blinatumomab responsive.

3. Did the authors observe any differences between CD8 and CD4 cells in the experiments with primary T cells.

→In these experiments, we have only worked with a mix of CD8 and CD4 T cell populations. With that in mind, we have not observed any specific differences between engineering CD8 and CD4 T cells. Both CD4 and CD8 T cells seem to be receptive to the mutant receptor and can express a functional TCR/CD3 complex on their surface.

Rebuttal 2

Reviewer 1

1. The authors claim their approach as ‘structure-based’ to identify a mutant that could decouple signal transmission from antigen engagement. However, they did not answer the question that why FEQWT is the best mutant for this approach? At the least they should screen for more mutants to identify the best possible mutant.

→ We appreciate R1’s comment. However, to capture the sequence space of mutations in a robust manner, we have used a library approach as described in Fig. 3 (pg. 9) and Supp. Fig. 6 (pg. 6 and 7). This data shows that we have actually screened over 400 variants across three TCRs, and for the selected variants, we further tested their reactivity with both peptide and blinatumomab.

This has now been more clearly stated in the manuscript (pg.8, line 235).

2. It is hard to envision the universality of this approach as the same mutation on 2 other TCRs resulted in poor expression and blinatumomab activation.

→ We agree with R1 and have now clarified that the mutated variant (FEQWT) is not universal across TCRs (pg.8, line 228). Since this mutation is not compatible for all TCRs, we decided to screen across different clinically relevant TCRs and were able to identify AED variants with decoupled properties in each TCR, as depicted in Fig. 3 (pg.9). We would like to point that a potential therapeutic product would consist of T cells that are all engineered and purified to have the same specificity: e.g. AED_{DMF5 01}. The mono-specific T cell product would not have reactivity to the cognate peptide and still have full functionality when activated with a bispecific antibody (e.g. blinatumomab).

3. Further, no mechanical details about the workings of this mutation through MD and modeling are provided. The work is more suited to core immunology journals.

→ We thank R1 for the advice to perform MD and modeling, which we have now performed. In our analysis, we found differences in the root-mean-square fluctuation (RMSF) between DMF5 TCR and AED_{DMF5 01} across all chains except CD3z. Most observable were the differences in the lipid-bilayer positioning of CD3ε and CD3ε’ chains where multiple fluctuations between WT DMF5 TCR and AED TCR were higher than 50%. These findings give additional information on how FEQWT mutation in the J region of the alpha chain can have a profound effect on the distal components of the TCR/CD3 complex and how AED mutations have the potential to substantially attenuate the activation signal induced by peptide-HLA binding.

This MD analysis is included in the manuscript (pg.7, line 197-204, and Supp. Fig. 4a,b).

Reviewer 2

The authors addressed most of my queries. I have two points remaining:

1. Answer to 1c should be incorporated into the manuscript like the preceding points.

→ We have now included R2's suggestion in the manuscript (pg. 1, line 56-59).

2. Measuring the level of target cell killing would be very important for this study. Different effector functions (proliferation, cytokine secretion, cytolysis) require different levels of TCR stimulation, and avoiding unwanted tissue toxicity is very important for this conceptual study. These assays are easy to do.

→ We have now added the cytolysis measurement to the manuscript. As stated in the study from Bossi et al. (doi: 10.4161/onci.26840), the physiological expression of peptides presented on HLA range between 10-150 peptide molecules on the cell surface, which corresponds to 0.1-10 ng/mL of pulsed peptide in an in vitro assay. Using such physiologically relevant concentrations of the EAA peptide at a 1:1 E: T ratio (primary T cells: T2 cells), we observe no T2 cell killing by AED T cells, whereas the DMF5 T cells completely eradicated peptide-pulsed T2 cells. At the 10 ng/mL concentration of AAG peptide, we observe T2 cell killing, however, it should be noted that AAG is the peptide predominantly expressed on melanoma tumor cells (doi:10.1002/eji.201948085). For clinical application, additional pre-clinical development and testing would be required to confirm the acceptable safety profile of the AED_{DMF5 01} variant.

Further, we performed cytolysis studies with blinatumomab. We demonstrate that the E: T ratio determines the efficacy of blinatumomab as the low E: T ratio (1:10) showed very limited cytolysis across T cells (non-engineered, DMF5 TCR and AED_{DMF5 01}). However, at 1:1 E: T, AED_{DMF5 01} T cells eradicate Raji tumor cells, similar to the non-engineered donor T cells. Together, this data, in combination with proliferation and cytokine expression, demonstrate the drastically reduced responsiveness of AED_{DMF5 01} T cells to cognate antigen compared to DMF5 TCR, while AED_{DMF5 01} T cells have preserved responsiveness to bispecific antibody stimulation (blinatumomab).

This additional analysis is now described in the new Supp. Fig. 8e and 8f, revised text in the Results and Discussion (pg.12, lines 299-304; pg.13, lines 329-333 and pg. 17, lines 420-423).